# When is More Better?
# Efficient and Adaptive Modality Acquisition in Multimodal Learning

## Abstract

Multimodal machine learning can improve flexibility, performance, and robustness, but incorporating many modalities increases acquisition costs and system complexity. This motivates *adaptive modality acquisition* (AMA), where a subset of the most informative modalities is selected *before observation* to balance predictive performance against cost. Prior work has largely focused on population-level acquisition, selecting a fixed subset of modalities that performs well on average. In this work, we instead adapt modality acquisition *per sample*, which is critical in settings such as healthcare where the value and cost of additional tests depend on the specific patient, improving efficiency while naturally supporting heterogeneous acquisition costs. This setting leads to the problem of *multi-stage subset selection with unobserved items and heterogeneous costs*, which poses challenges in both uncertainty and scalability. Our key idea is to learn a compositional energy-based value function that scores candidate modality subsets for their expected contribution to downstream prediction. We implement this through *recursive value functions* that estimate the value of acquiring any subset of modalities conditioned on the currently observed modalities, allowing the same model to be applied iteratively as new modalities are acquired. Our key contributions are: **(1)** learning this recursive value function as an *energy-based model*; **(2)** designing and characterizing suitable value functions for this setting, with a selection rule based on the *model confusion rate* (MCR; probability that added modalities flip a correct prediction); and **(3)** showing that, under a natural submodularity assumption on modality value, the acquisition objective can be optimized efficiently via submodular optimization. This framework yields scalable training and inference algorithms, collectively referred to as EFFICIENT ADAPTIVE MODALITY ACQUISITION (EAMA), that scale linearly in the number of modalities. Across multiple real-world multimodal datasets with up to $M = 15$ modalities, EAMA achieves up to an $8\times$ improvement in balancing accuracy gains against acquisition costs relative to baseline methods. In some cases, EAMA is able to *do more with less*, improving accuracy while using only 27.4% of the available modalities on average.

## 1 Introduction

A growing body of literature has provided significant empirical evidence for the benefit of multiple data modalities for improving the predictive performance of machine learning (ML) models (Soenksen et al., 2022; Bapna et al., 2022; Zhang et al., 2025; Gabeff et al., 2023; Liang et al., 2023b). Additionally, theoretical analysis has yielded intuitive results demonstrating that multimodal learning has the ability to reduce population risk (Huang et al., 2021) and improve prediction performance from an information-theoretic perspective (Liang et al., 2023a; He et al., 2024). Despite its potential, multimodal learning comes with the inherent cost of acquiring and processing multiple data sources (Gao et al., 2020; Zong et al., 2024). These costs can come in the form of money, time, energy, and even physical health in cases such as medical imaging (Luccioni et al., 2024; Warner et al., 2024). It is natural to consider the decision making problem of selecting modalities to acquire, before they have been observed, in order to maximize performance and

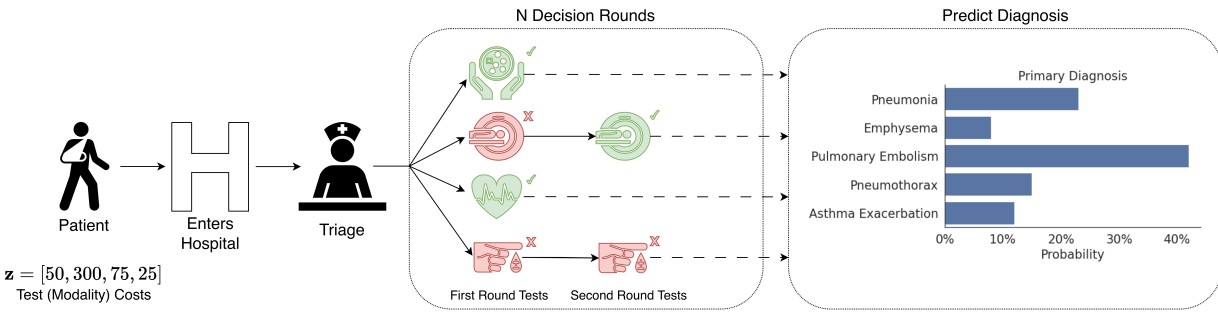

Figure 1: The adaptive modality acquisition (AMA) problem in a clinical setting. AMA seeks to select subsets of unobserved modalities over multiple stages to maximize predictive performance while minimizing acquisition costs. Here, modalities correspond to medical tests. Given a patient's context (e.g., chief complaint and vitals) and patient-specific costs (e.g., monetary cost, health risk, and time), the task is to decide which unobserved tests to order simultaneously, before their outcomes are known, to obtain sufficient information for accurate diagnosis at minimal cost.

minimize costs. We call this the *adaptive modality acquisition* (AMA) problem, where subsets of unobserved modalities can be acquired over multiple time steps or stages, subject to heterogeneous acquisition costs, in order to maximize the performance of a predictive model. Such problems occur in many medical, ecological, and financial applications (Bernardino et al., 2022; Hulkund et al., 2025; Farimani et al., 2024). For example, consider the problem of selecting medical tests to administer to patients admitted to a hospital's emergency department (ED). At admission, the hospital collects some basic information about the patient, such as their demographics, vitals, and chief complaint. We then want to select a set of medical tests to give to the patient in order to narrow our diagnosis. The costs of each medical test might be specific to that patient, depending on insurance, test availability, urgency, or frailty of the patient (Wray et al., 2023; Whaley et al., 2025; Shen & Moss, 2022). After the selected test results are acquired, we must again decide whether to administer additional tests, or conclude testing and diagnose, treat and discharge the patient. In this work, we would assume a multimodal classifier is used to predict the final diagnosis using the acquired modalities. Settings like this demonstrate the necessity of subset selection, as opposed to acquiring one modality at a time, as well as adapting to heterogeneous costs.

In this work, we address the challenging problem of *multi-stage subset selection with unobserved, high-dimensional items and heterogeneous acquisition costs*. Adaptive modality acquisition (AMA) presents challenges arising from both uncertainty and scale: at each stage, there is uncertainty in the values of unobserved modalities as well as in how the predictive model's output will change after acquiring additional observations. Moreover, the underlying subset selection problem is NP-hard and scales exponentially in the number of modalities, posing difficulties not only at inference time but also during training, where the value of acquiring any subset of unobserved modalities must be estimated given any subset of observed modalities. To address these challenges, we focus on the design and optimization of *recursive value functions* for AMA, that estimate the value of acquiring any subset of modalities conditioned on the currently observed modalities, allowing the same model to be applied iteratively as new modalities are acquired. We frame this problem through the lens of energy-based learning (LeCun et al., 2006), enabling us to learn a single, scalable value model that estimates the value of acquiring any subset of unobserved modalities conditioned on the currently observed set. By learning latent representations for each modality, we design value functions that can reason over arbitrary modality combinations in a sample-efficient manner while scaling linearly in the number of modalities. Assuming that modality value is submodular, we further observe that selecting modalities to maximize value minus acquisition cost corresponds to a non-monotonic submodular optimization problem, for which linear-time approximation algorithms with theoretical guarantees exist. Finally, we define and characterize a family of value functions and derive a simple rule for selecting the appropriate one for a given task. This rule is based on a metric we term the *model confusion rate*, which measures how often a classifier is confused by additional modalities and plays a central role in determining when more information is beneficial. Together, these insights yield efficient training and inference algorithms, which we refer to as

| Property | FS | EMS | MDP | MORL | CB | EAMA (Ours) |
|---|---|---|---|---|---|---|
| Multi-stage | ✗ | ✗ | ✓ | ✓ | ✗ | ✓ |
| Context-aware | ✗ | ✗ | ✓ | ✓ | ✓ | ✓ |
| Subset selection | ✓ | ✓ | ✗ | ✗ | ✓ | ✓ |
| Heterogeneous costs | ✗ | ✗ | ✗ | ✓ | ✓ | ✓ |
| High-dimensional features | ✗ | ✓ | ✓ | ✓ | ✓ | ✓ |
| Modality Scaling | $O(M^d)^\dagger$ | $O(M^2)$ | $O(2^M)$ | $O(2^M)$ | $O(2^M)$ | $O(M)$ |

$\dagger$ Polynomial degree $d$ varies depending on the feature selection method.

Table 1: Comparison of EAMA and Feature Selection (FS) (Chandrashekar & Sahin, 2014), Efficient Modality Selection (EMS) He et al. (2024), Markov Decision Processes (MDP) (Bernardino et al., 2022), Multi-Objective RL (MORL) (Yang et al., 2019), and Contextual Bandits (CB) (Collier & Llorens, 2018).

EFFICIENT ADAPTIVE MODALITY ACQUISITION (EAMA), for finding high-quality approximate solutions to the AMA problem. Across multiple real-world multimodal datasets with up to $M = 15$ modalities, EAMA achieves up to an $8\times$ improvement in balancing accuracy gains against acquisition costs compared to baseline methods, and in some cases is able to *do more with less*, improving accuracy while using only 27.4% of the available modalities on average.

## 2   Related Work

The AMA problem is closely related to ideas from reinforcement learning, bandits, feature selection, and active learning. We discuss how these techniques differ from the structure of the AMA problem below. Critically, these other techniques either face scaling challenges, scaling exponentially in the number of modalities, or are unable to adapt to heterogeneous acquisition costs. In Table 1, we compare the most closely-related techniques to our EAMA algorithm along key properties of the AMA problem.

**Reinforcement Learning.**  An alternative to our approach for AMA is to consider the problem in a reinforcement learning (RL) framework. While standard methods such as PPO (Schulman et al., 2017) could be used, Bernardino et al. (2022) and Wang et al. (2015) observed that this general learning problem can be formulated as a Partially-Observable Markov Decision Process (POMDP) with a special structure that can be solved explicitly. However, these approaches scale exponentially in the number of modalities, and cannot handle heterogeneous acquisition costs. In order to incorporate heterogeneous acquisition costs in an RL framework, one would have to turn to multi-objective RL (Yang et al., 2019), but these techniques do not scale well as they require training over the entire space of costs, which would grow exponentially in $M$. In contrast, our algorithm, EAMA, scales linearly in $M$ while still considering interaction effects between modalities.

**Contextual Bandits.**  In the single-stage setting, AMA is similar to an offline version of the contextual multi-arm bandit (CB) learning problem (Lu et al., 2010; Collier & Llorens, 2018). However, CB algorithms struggle to scale to high-dimensional discrete action spaces (Saito & Joachims, 2022), and do not offer solutions to the multi-stage setting in which we consider the AMA problem.

**Modality and Feature Selection.** Most previous work in the space of feature selection (Chandrashekar & Sahin, 2014; Pudil et al., 1994) and modality selection (He et al., 2024) focus on the population-level, selecting a single subset of features or modalities to be used at inference for all samples. Adaptive feature acquisition (Janisch et al., 2019) and modality acquisition (Bernardino et al., 2022) have considered only the multi-stage setting and proposed RL or dynamic programming solutions. Adaptive feature acquisition, in which each feature is a single scalar value, differs significantly from our setting where each modality is a high-dimensional vector. The proposed method in Bernardino et al. (2022) scales exponentially in $M$, and neither method can incorporate varying acquisition costs.

**Active Learning.** The AMA problem is also related to acquisition problems in active learning. Recent work in active learning has focused on learning acquisition functions, balancing information gain with acquisition costs (Fang et al., 2017; Liu et al., 2018; Haussmann et al., 2019; Schrum et al., 2020; Taguchi et al.,

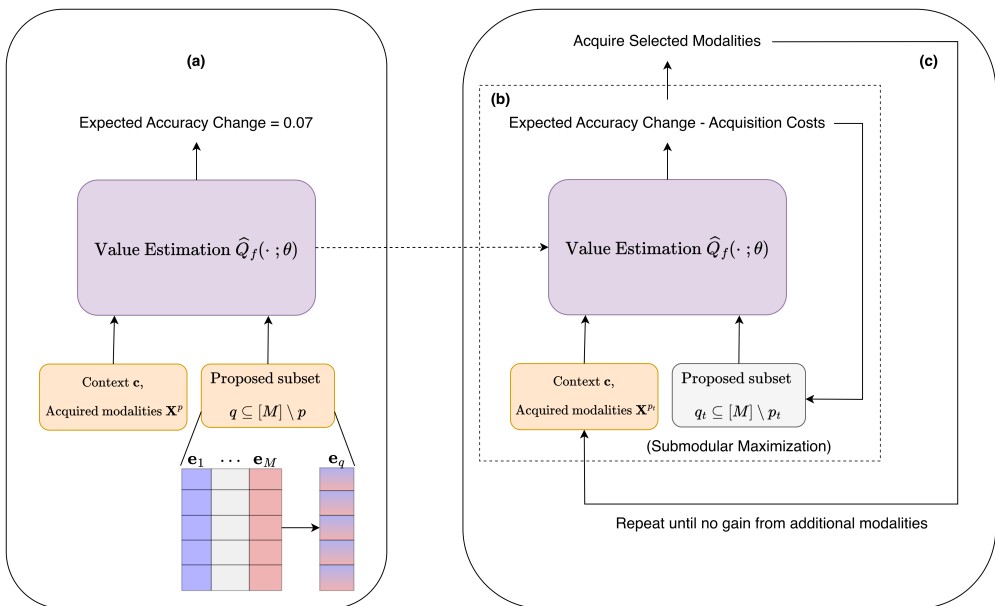

Figure 2: An overview of EFFICIENT ADAPTIVE MODALITY ACQUISITION (EAMA). **a)** We learn a value model $\widehat{Q}_\theta$ by generating views of the samples and subsets of modalities to acquire. We train the model to predict the true value of acquiring the selected subset of modalities. **b)** Using the learned value model $\widehat{Q}_\theta$, at inference, we optimize the selected subset $q_t$ to maximize the estimated value minus acquisition costs. **c)** After optimizing $q_t$, we acquire the selected modalities and repeat (b) until either $p_t = [M]$ or the maximum objective value is no longer positive, at which point we predict with $\hat{y} = f(\mathbf{c}, \mathbf{X}^{p_t})$.

2021). While the setup is similar to our work and the AMA problem, they fundamentally differ in the problem being solved (selecting modalities versus selecting new samples) and the structure of the problem. For AMA, the acquisition problem must be solved for every sample, potentially across multiple stages per sample. In contrast, the active learning problem must be solved per task. The AMA problem operates on partial observations of individual samples, whereas active learning methods assume access to a training set of fully observed samples.

## 3 Adaptive Modality Acquisition

In this section, we formally define the AMA problem and develop our EAMA methodology. We begin by posing the problem in Section 3.1, decomposing the overall problem into two stages: learning a value model to estimate the expected value of acquiring additional modalities, and then optimizing this value model, given acquisition costs. In Section 3.2, we introduce our efficient approach to solving modality acquisition via submodular estimation. Then, in Section 3.3, we describe our methodology for efficiently learning value models over pairs of observed modalities and candidate subsets of unobserved modalities. Finally, in Section 3.4, we introduce our value function definitions, based on accuracy change, and provide theoretical insights that guide the selection of value function for a specific application.

### 3.1 Problem Definition

To organize our analysis, let us assume access to a dataset $\{\mathbf{X}_i, y_i\}_{i=1}^N$ of $N$ samples from the joint distribution $(\mathbf{X}, y) \sim \mathcal{D}$, where $\mathbf{X}_i = (\mathbf{c}_i, \mathbf{x}_i^1, \ldots, \mathbf{x}_i^M)$ contains the context modality and complete representation of the sample across all $M$ modalities and $y \in [K]$ is the associated label. We assume measurements $\mathbf{x}^j \in \mathbb{R}^{n_j}$ from each modality $j \in [M]$ and context modality $\mathbf{c} \in \mathbb{R}^{n_c}$, such that each modality can have its own

dimensionality. We use the term *context modality* to refer to the information we can assume access to at the start of the modality acquisition process. For example, in a clinical setting where each sample represent a hospital patient, the context modality $\mathbf{c}_i$ might be demographic information, basic vitals, or a written triage note, and the modalities $\mathbf{x}_i^1, \ldots, \mathbf{x}_i^M$ could be various lab tests, clinical notes, or images (e.g. CT scans, X-rays). Given a subset of modalities $p \subseteq [M]$, we define a **view** $\mathbf{X}_i^p = (\mathbf{x}_i^j)_{j \in p}$ of sample $\mathbf{X}_i$ as the corresponding subset of data from those modalities for that sample. We introduce this definition both for convenience and to illustrate that in the context of multimodal learning, we can observe many different, potentially incomplete, views of the same sample. For the AMA problem specifically, at any point in time, we will only have access to a subset of observed modalities corresponding to some view $\mathbf{X}^p$, while the target, $y$, and all other views, $\mathbf{X}^q$ for $q \subseteq [M] \setminus p$, remain unobserved. For notational simplicity, we suppress the sample index $i$ and write $\mathbf{X}_p$ to denote the view for a fixed instance. Following the notation of Bernardino et al. (2022), we define the full state space as the disjoint union of the cartesian product of each subset of modalities along with the context,

$$\mathcal{S} = \dot{\bigcup_{p \subseteq [M]}} \left( \prod_{j \in p \cup \{c\}} \mathbb{R}^{n_j} \right). \tag{1}$$

That is, every $s \in \mathcal{S}$ represents a view of a sample and its context. Importantly, we assume access to a model class $\mathcal{F}$ of $K$-class multimodal classification models $f : \mathcal{S} \to \mathbb{R}^K$. We assume the models in this class can take as input any view of a sample, along with the associated context. With recent advances in large language models (LLMs), such models are not only readily available, but often yield state-of-the-art performance for a large variety of learning tasks. Further, with many developed techniques to represent tabular data sources and image data as text (Carballo et al., 2023; Sellergren et al., 2025), the most powerful text-only LLMs can now process complex multimodal data. These developments vastly simplify the complexity of AMA, where previously many models would have to be created for every possible combinations of modalities (Bernardino et al., 2022), or complex schemes to represent missingness would have to be employed (Wu et al., 2024).

We now formalize the notion of measuring the value of modality acquisition. Given a classification model $f \in \mathcal{F}$, we must define a value function $Q_f : \mathcal{S} \times \mathcal{S} \times [K] \to \mathbb{R}$ which quantifies the value of acquiring any view of a sample, given an existing view and the ground truth label. This value function represents the true value for acquiring additional modalities, given complete information about all modalities and the associated label. For a given sample $(\mathbf{X}, y) \sim \mathcal{D}$, a subset of acquired modalities $p \subset [M]$, and a subset $q \subseteq [M] \setminus p$ of the remaining modalities, we denote the value function as $Q_f(\mathbf{c}, \mathbf{X}^p, \mathbf{X}^q, y)$. Our goal is to estimate the expected value of acquiring view $\mathbf{X}^q$, which we define as

$$\widehat{Q}_f(\mathbf{c}, \mathbf{X}^p, q) \stackrel{\text{def}}{=} \mathbb{E}_{y, \mathbf{X}^q | \mathbf{c}, \mathbf{X}^p}[Q_f(\mathbf{c}, \mathbf{X}^p, \mathbf{X}^q, y)], \tag{2}$$

where the true label $y$ and view $\mathbf{X}^q$ are sampled from the conditional distribution induced by the joint distribution, $\mathcal{D}$, on the context and acquired modalities. Note that these definitions cover the general AMA problem, as setting $p = \emptyset$ recovers the first-stage problem of selecting a subset $q \subseteq [M]$ given only the context. For a specific value function $Q_f$, we want to learn $\widehat{Q}_f : \mathcal{S} \times \{0, 1\}^M \to \mathbb{R}$, which maps any state-action pair to the expected value of that action. This is the core learning problem for our EAMA approach.

Assuming we have access to $\widehat{Q}_f$, or we have learned a value model to approximate it, we solve the AMA problem as follows. First, suppose we have an instance-specific cost-vector $\mathbf{z} = (z_1, \ldots, z_M) \in \mathbb{R}_+^M$. Then at any stage $t$, with acquired modalities $p_t \subset [M]$, we want to find

$$q_t^* = \underset{q_t \subseteq [M] \setminus p_t}{\arg\max} \, \widehat{Q}_f(\mathbf{c}, \mathbf{X}^{p_t}, q_t) - \lambda \sum_{i \in q_t} z_i, \tag{3}$$

the subset of unobserved modalities that maximizes the trade-off between expected value gain and acquisition costs. We make the reasonable simplifying assumption that acquisition costs are additive. This formulation greedily selects the best subset of modalities in expectation at any time step $t$, conditioned on the information already acquired. At the end of each stage, the modalities $q_t^*$ are acquired, and we define $p_{t+1} = p_t \cup q_t^*$ as the new set of acquired modalities. This subset selection problem is NP-Hard in general, scaling exponentially in the number of modalities. However, this formulation decouples value estimation and acquisition costs such that, given we can efficiently find a solution to Equation 3, we can gracefully handle heterogeneous acquisition

costs. Viewing the AMA problem through the lens of *energy-based learning* (LeCun et al., 2006), where a model learns a scalar compatibility (or energy) function over input pairs, we decompose the problem into two components. First, we learn a value model $\widehat{Q}_f(\cdot\ ;\ \theta)$ defined over the space $\mathcal{S} \times \{0,1\}^M$ of observed modalities and candidate subsets of modalities to acquire. Second, at each stage we solve the optimization problem in Equation 3 to select the subset $q_t \in \{0,1\}^M$ that maximizes the expected value given the current view $(\mathbf{c}, \mathbf{X}^{p_t}) \in \mathcal{S}$. In Section 3.3, we describe our methodology for efficiently learning value models, addressing the first stage in the EAMA approach. To efficiently find approximate solutions to the optimization problem in Equation 3, we can rely on a common property of multimodal learning.

### 3.2 Modality Acquisition via Submodular Optimization

At inference, during each stage, we must solve the optimization problem posed in Equation 3. To make this approach practical for real-world use, we must develop an efficient method to find high-quality approximate solutions to this optimization problem. To accomplish this, we rely on the observation that incorporating additional modalities to a multimodal learning problem *exhibits diminishing marginal returns*, which has been observed empirically (Soenksen et al., 2022) and established from an information-theoretic perspective (Liang et al., 2023a; He et al., 2024; Huang et al., 2021). More specifically, we make the assumption that $\widehat{Q}_f$ is *submodular* in the set $q$ of modalities to acquire, which we formalize as follows.

**Assumption 3.1.** *(Submodular expected value) For a fixed state $(\mathbf{c}, \mathbf{X}^p) \in \mathcal{S}$, the expected value function $\widehat{Q}_f$ is submodular. That is, $\forall q_1 \subseteq q_2 \subseteq [M] \setminus p, j \in [M] \setminus q_2 \cup p, \widehat{Q}_f(\mathbf{c}, \mathbf{X}^p, q_2 \cup \{j\}) - \widehat{Q}_f(\mathbf{c}, \mathbf{X}^p, q_2) \leq \widehat{Q}_f(\mathbf{c}, \mathbf{X}^p, q_1 \cup \{j\}) - \widehat{Q}_f(\mathbf{c}, \mathbf{X}^p, q_1).$*

This assumption, which has been empirically and theoretically validated, provides crucial structure for optimizing the objective in Equation 3. To validate this assumption empirically, we test the second-order submodularity condition for all of our experiments and report the full results in Appendix C. Notably, we find that across our experiments, there are zero statistically significant violations of this submodularity test, and the violation rate ranges from $0.35\% - 4.5\%$, with an average violation rate of $1.98\%$ across experiments. For convenience, let us define the objective function as $\phi(q\ ;\ \mathbf{c}, \mathbf{X}^p, \mathbf{z}, \lambda) = \widehat{Q}_f(\mathbf{c}, \mathbf{X}^p, q) - \lambda \sum_{i \in q} z_i$, or $\phi(q)$ for a fixed state and cost vector. Assuming $\widehat{Q}_f$ is submodular, we can characterize $\phi(q)$ in Lemma 1.

**Lemma 1.** *Given Assumption 3.1, the objective $\phi(q)$ is a non-monotonic submodular function.*

The proof of Lemma 1 is given in Appendix A and relies only the acquisition costs being additive. With this result, we can now view the optimization problem in Equation 3 as an *unconstrained, non-monotonic submodular maximization problem*. For this class of problems, there exists a randomized $\frac{1}{2}$-approximation algorithm, RandomizedUSM (Buchbinder et al., 2012) (RUSM), which runs in $O(M)$ time, requiring a total of $2M + 2$ evaluations of $\phi(\cdot)$. For applications where a randomized algorithm is not acceptable, there exists a deterministic algorithm with the same $\frac{1}{2}$-approximation guarantee, introduced by Buchbinder & Feldman (2015), which runs in $O(M^2)$ time. We summarize the randomized algorithm in the context of AMA in Algorithm 3. Under the stated assumptions, Algorithm 3 provides tight theoretical performance guarantees. At a high-level, RUSM works by tracking two sets of modalities, one initialized as the empty set, the other as the set of all unobserved modalities. At each iteration, the algorithm measures the value of adding a given modality to the first set, or removing it from the second, before deciding which action to take. After iterating over all unobserved modalities, the two sets are equal, representing the final subset selected.

While the RUSM algorithm has the only performance guarantees in this setting, it is a pessimistic algorithm, designed to avoid the worst-case scenarios. Alternatively, we can use the standard greedy submodular maximization algorithm (Nemhauser et al., 1978), a more optimistic algorithm that, unlike RUSM, will not eliminate modalities from consideration prematurely. However, the greedy algorithm has no performance guarantees in this setting. Therefore, we take a hybrid approach, running both algorithms and returning whichever result maximizes the objective function. This hybrid algorithm scales in $O(M^2)$, retains the performance guarantees of RUSM, and mitigates the pessimistic nature of the RUSM algorithm. We provide further details and an ablation study over these optimization algorithms in Appendix B. Collectively, these techniques make up our EAMA methodology, summarized in Algorithm 1.

---

**Algorithm 1** Efficient Adaptive Modality Acquisition

---

**Require:** Context modality $\mathbf{c}$, value model $\widehat{Q}_f(\cdot\,;\theta)$, cost vector $\mathbf{z}$, cost parameter $\lambda$
$\quad p_0 \leftarrow \emptyset,\ q_0 \leftarrow \emptyset,\ t \leftarrow 0$
$\quad$ **while** $|p_t| < M$ and $\phi(q_t) \geq 0$ **do**
$\quad\quad q_{t+1}^1 \leftarrow \mathrm{RUSM}(\mathbf{c}, \mathbf{X}^{p_t}, \widehat{Q}_f, \mathbf{z}, \lambda)$ $\hfill \triangleright$ Algorithm 3 (Buchbinder et al., 2012).
$\quad\quad q_{t+1}^2 \leftarrow \mathrm{Greedy}(\mathbf{c}, \mathbf{X}^{p_t}, \widehat{Q}_f, \mathbf{z}, \lambda)$ $\hfill \triangleright$ (Nemhauser et al., 1978).
$\quad\quad$ **if** $\phi(q_{t+1}^1) > \phi(q_{t+1}^2)$ **then**
$\quad\quad\quad q_{t+1} \leftarrow q_{t+1}^1$
$\quad\quad$ **else**
$\quad\quad\quad q_{t+1} \leftarrow q_{t+1}^2$
$\quad\quad$ **end if**
$\quad\quad$ **if** $\phi(q_{t+1}) \geq 0$ **then**
$\quad\quad\quad p_{t+1} \leftarrow p_t \cup q_{t+1}$
$\quad\quad\quad \mathbf{X}^{p_{t+1}} \leftarrow \mathbf{X}^{p_t} \cup \mathbf{X}^{q_{t+1}}$ $\hfill \triangleright$ Acquire modalities $\mathbf{X}^{q_{t+1}}$.
$\quad\quad$ **else**
$\quad\quad\quad p_{t+1} \leftarrow p_t,\ \mathbf{X}^{p_{t+1}} \leftarrow \mathbf{X}^{p_t}$
$\quad\quad$ **end if**
$\quad\quad t \leftarrow t + 1$
$\quad$ **end while**
$\quad$ **Return** Final state $(\mathbf{c}, \mathbf{X}^{p_t})$

---

### 3.3 Efficiently Learning Value Functions for EAMA

In the previous section, we defined our EAMA algorithm for efficiently solving the AMA problem at inference, given access to an existing value model. In this section, we discuss our approach to efficiently learning value models for the AMA problem, which is illustrated in Figure 2. Specifically, we want to learn the value function $\widehat{Q}_f : \mathcal{S} \times \{0, 1\}^M \to \mathcal{V}$ with a neural network, denoted by $\widehat{Q}_f(\cdot\,;\,\theta)$, which we refer to as the *value model*, where $\theta$ parameterizes the network. We denote the value model by $\widehat{Q}_f(\cdot\,;\,\theta)$ to mirror the notation of the expected value function $\widehat{Q}_f$, since we will use the value model to estimate the expected value of acquiring additional modalities. The specific architecture of the network itself depends on the application and modalities in use, and we provide details of the implementations for our experiments in Section 4. However, the way we encode subsets of modalities is the same across our experiments. For each modality $i \in [M]$, we construct a learnable embedding $\mathbf{e}_i \in \mathbb{R}^d$, where $d \ll 2^M$ and we constrain $||\mathbf{e}_i||_2 = 1$. For any subset $p \subseteq [M]$, we encode the subset using the mean of the individual modality embeddings, $\mathbf{e}_p = \frac{1}{|p|}\sum_{i \in p} \mathbf{e}_i$. This approach efficiently encodes subsets of modalities, allowing the model to learn relationships between modalities and use the magnitude of the embedding, $||\mathbf{e}_p||_2$, to encode the size of the subset, which we investigate in Appendix C. In practice, we encode both the subset of acquired modalities *and* the selected subset of modalities to acquire. Therefore, for acquired modalities $p$ and selected subset $q \subseteq [M] \setminus p$ to acquire, the input to $\widehat{Q}_f(\cdot\,;\,\theta)$ would be $(\mathbf{c}, \mathbf{X}^p, \mathbf{e}_p, \mathbf{e}_q)$, efficiently representing the state and action.

To train the model, for each sample $(\mathbf{X}, y)$, we randomly sample subsets $p \subset [M]$ and $q \subseteq [M] \setminus q$ and train the model to predict $Q_f(\mathbf{c}, \mathbf{X}^p, \mathbf{X}^q, y)$, which is summarized in Algorithm 2. Critically, we sample the number of observed modalities $k_1$ uniformly over $[M - 1]$, then sample $p$ uniformly over all $[M]$ choose $k_1$ subsets of modalities. We then repeat this sampling process for the selected subset $q \subset [M] \setminus p$ of the unobserved modalities. This method helps ensure that the value model is trained on varying sizes of observed and selected subsets of modalities even for $M$ large. In most practical applications, we will start with $p_0 = \emptyset$, and only end up selecting a total of $k << M$ modalities to acquire, so we want to make sure the value model performs well when $p$ is small. In other scenarios with lower acquisition costs, we might select the majority of the modalities in most cases, so sampling both tails for the size of $p$ can be important. In contrast, were we to sample $p \sim \mathrm{Binom}(M, 0.5)$, including each modality in $p$ at random as a coin-flip, the odds of $p$ being close to 0 or $M - 1$ becomes vanishingly small for large $M$. We note that this distribution could be further

---

**Algorithm 2** Training Value Models for AMA

---

**Require:** Dataset $D = \{\mathbf{X}_i, y_i\}_{i=1}^N$, multimodal classifier $f$, initialized value model $\widehat{Q}_f(\,\cdot\;;\,\theta)$
  **while** loss not converged **do**
    **for** $(\mathbf{X}_i, y_i) \in D$ **do**
      Sample $k_1 \sim \mathrm{Unif}(0, M-1)$
      Sample $k_2 \sim \mathrm{Unif}(1, M-k_1)$
      Sample $p \sim \mathrm{Unif}\left(\binom{[M]}{k_1}\right)$
      Sample $q \sim \mathrm{Unif}\left(\binom{[M]\setminus p}{k_2}\right)$
      Take gradient step on $\nabla_\theta \mathcal{L}(\widehat{Q}_f(\mathbf{c}, \mathbf{X}^p, q\;;\,\theta), Q_f(\mathbf{c}, \mathbf{X}^p, \mathbf{X}^q, y))$
    **end for**
  **end while**

---

adjusted in practice given domain knowledge of the task and acquisition costs. Combining Algorithms 1 and 2 gives us our EAMA pipeline, providing a simple, scalable method for solving the AMA problem.

### 3.4 Value Functions for AMA

The choice of value function is a critical design decision that strongly influences the quality of the EAMA solution to the AMA problem. Since we focus on classification tasks, we design value functions that measure the change in accuracy from acquiring additional modalities. We base our formulation on accuracy for two primary reasons. First, accuracy change is an interpretable metric that directly reflects improvements in outcomes, making it straightforward to calibrate value gains against acquisition costs in practical applications. If costs are given in a known unit, such as dollars, then practitioners can understand the cost parameter $\lambda$ as the exchange rate between cost and expected accuracy change. Second, accuracy change can be measured pointwise at the level of individual inputs, whereas most alternative metrics, such as ROC-AUC or F1-score, are defined only at the population level. Accuracy also naturally extends to multiclass classification, and all value functions and analysis in this work generalize directly to top-$k$ accuracy. We now introduce the main value function used in this work and then decompose it, yielding a simple rule for deciding when to use this value function or a related, simpler alternative.

For convenience, for any $f \in \mathcal{F}$, we define $f_c(\cdot) \stackrel{\text{def}}{=} \arg\max_{y \in [K]} f(\cdot)_k$. We can then define the *accuracy change* value function as

$$Q_f^A(\mathbf{c}, \mathbf{X}^p, \mathbf{X}^q, y) = \mathbf{1}\{f_c(\mathbf{c}, \mathbf{X}^{p \cup q}) = y\} - \mathbf{1}\{f_c(\mathbf{c}, \mathbf{X}^p) = y\}. \tag{4}$$

This value function encodes three states: if $Q_f^A(\cdot) = 1$, then adding view $\mathbf{X}^q$ causes the model to flip to predicting the true label, if $Q_f^A(\cdot) = 0$, then adding $\mathbf{X}^q$ does not change the prediction, and if $Q_f^A(\cdot) = -1$, adding $\mathbf{X}^q$ has confused the model, causing it to flip to predicting an incorrect label. This value function has multiple beneficial properties. First, taking expectations, we find

$$\widehat{Q}_f^A(\mathbf{c}, \mathbf{X}^p, \mathbf{X}^q, y) = P(f_c(\mathbf{c}, \mathbf{X}^{p \cup q}) = y \mid \mathbf{c}, \mathbf{X}^p) - P(f_c(\mathbf{c}, \mathbf{X}^p) = y \mid \mathbf{c}, \mathbf{X}^p),$$

which is exactly the change in accuracy by acquiring $\mathbf{X}^q$ given the current observations. As we show in Lemma 3, if we train a value model to predict $Q_f^A$ with cross-entropy loss, the optimal value model is exactly $\widehat{Q}_f^A$, yielding a continuous-valued estimate of the expected change in accuracy by acquiring a new subset of modalities. Additionally, $Q_f^A$ is a telescoping function, such that for any sequence of subsets $\emptyset = p_0 \subset p_1 \cdots \subset p_T \subseteq [M]$, we have $\sum_{t=1}^T Q_f^A(\mathbf{c}, \mathbf{X}^{p_{t-1}}, \mathbf{X}^{p_t \setminus p_{t-1}}, y) = Q_f^A(\mathbf{c}, \mathbf{X}^{p_0}, \mathbf{X}^{p_T}, y)$. This telescoping property is key in the context of multi-stage subset selection.

Simplifying one step further, we construct the *bit flip* value function, which we define as

$$Q_f^B(\mathbf{c}, \mathbf{X}^p, \mathbf{X}^q, y) = \mathbf{1}\{f_c(\mathbf{c}, \mathbf{X}^{p \cup q}) = y \;\wedge\; f_c(\mathbf{c}, \mathbf{X}^p) \neq y\}, \tag{5}$$

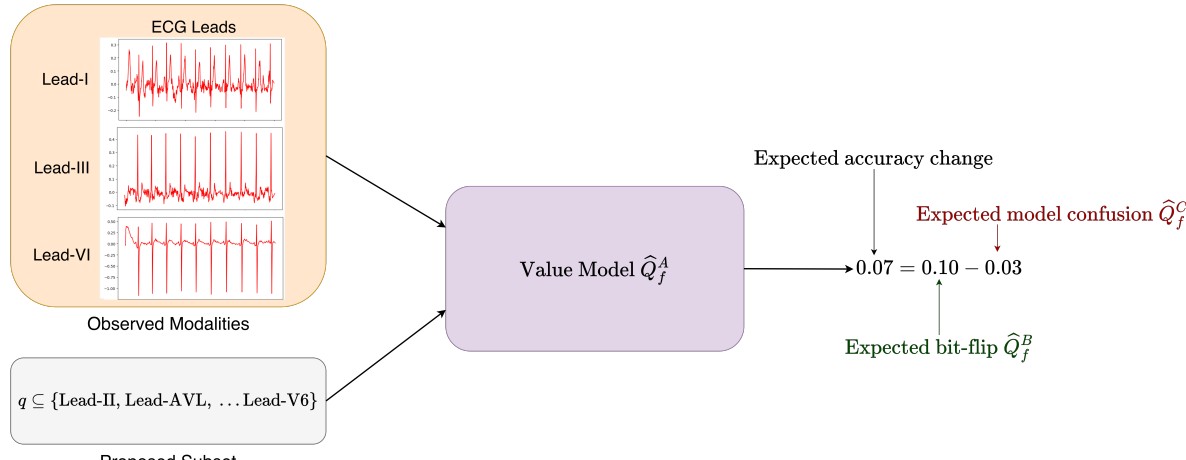

Figure 3: Based on the paradigm of energy-based learning, the value model takes the current state (e.g. observed ECG leads) and any subset of unobserved modalities as input, and outputs a scalar compatibility score. In this work, that score represents the expected change in accuracy, which can be decomposed into the probability of the classifier flipping from an incorrect to a correct prediction, minus the probability of the *classifier becoming confused* and flipping from a correct to an incorrect prediction.

which indicates whether adding view $\mathbf{X}^q$ flips the classifier's prediction from an incorrect class to the correct class. This value function limits the value space to $\mathcal{V} = \{0, 1\}$, such that learning $\widehat{Q}_f^B$ becomes a binary classification problem. This value function is directly connected to the accuracy change value function, which can be seen by rewriting $Q_f^A$ as follows,

$$Q_f^A(\mathbf{c}, \mathbf{X}^p, \mathbf{X}^q, y) = Q_f^B(\mathbf{c}, \mathbf{X}^p, \mathbf{X}^q, y) - \mathbf{1}\{f_c(\mathbf{c}, \mathbf{X}^{p\cup q}) \neq y \wedge f_c(\mathbf{c}, \mathbf{X}^p) = y\}.$$

From this relationship, we can see that the two value functions differ only by the model confusion term. We can define this confusion term as another value function, which we refer to as the *model confusion* value function, $Q_f^C(\mathbf{c}, \mathbf{X}^p, \mathbf{X}^q, y) = \mathbf{1}\{f_c(\mathbf{c}, \mathbf{X}^{p\cup q}) \neq y \wedge f_c(\mathbf{c}, \mathbf{X}^p) = y\}$. With these value functions defined, we can succinctly capture the relationship between them as

$$\underbrace{Q_f^A(\mathbf{c}, \mathbf{X}^p, \mathbf{X}^q, y)}_{\text{Accuracy Change}} = \underbrace{Q_f^B(\mathbf{c}, \mathbf{X}^p, \mathbf{X}^q, y)}_{\text{Bit-Flip}} - \underbrace{Q_f^C(\mathbf{c}, \mathbf{X}^p, \mathbf{X}^q, y)}_{\text{Model Confusion}}. \tag{6}$$

We provide further details of this derivation in Lemma 2. While the model confusion value function is not useful on its own for solving the AMA problem, we can see through Equation 6 that measuring $Q_f^C$ can help us determine whether to use $Q_f^A$ or $Q_f^B$ to solve the AMA problem. Specifically, if $E[Q_f^C(\cdot)] \approx 0$ over random subsets $p, q \subset [M]$, then $Q_f^A \approx Q_f^B$ and it is better to use $Q_f^B$ as our value function, as it is a simpler and more well-balanced learning problem. Alternatively, if the expected value of $Q_f^C$ is non-negligible, then using $Q_f^A$ is the better choice for solving the AMA problem. We refer to $E[Q_f^C(\cdot)]$ as the *model confusion rate* (MCR). The issue is compounded in the multi-stage setting of the AMA problem, as we can see for a sequence of subsets,

$$\underbrace{\sum_{t=1}^T Q_f^B\left(\mathbf{c}, \mathbf{X}^{p_{t-1}}, \mathbf{X}^{p_t \setminus p_{t-1}}, y\right)}_{\text{Cumulative Bit-Flip Value}} = \underbrace{Q_f^A(\mathbf{c}, \mathbf{X}^{p_0}, \mathbf{X}^{p_T}, y)}_{\text{Accuracy Change}} + \underbrace{\sum_{t=1}^T Q_f^C\left(\mathbf{c}, \mathbf{X}^{p_{t-1}}, \mathbf{X}^{p_t \setminus p_{t-1}}, y\right)}_{\text{Cumulative Model Confusion}}, \tag{7}$$

such that $Q_f^B$ will yield an optimistic measure of value gain, which accumulates over time steps, as it is unable to capture the possibility of the classifier becoming confused by additional information. In contrast, the key to $Q_f^A$ having the telescoping property lies in its ability to capture model confusion at each stage. In

Section 4, for each dataset, we estimate the model confusion rate empirically and show examples of where this condition clearly influences which value function is better suited for solving the AMA problem. In addition to these value functions, we also explored a more granular value function, which measures the value in terms of *information gain*, the change in the entropy of the classifier's predictions by acquiring new modalities. However, across datasets, the noisy, high-variance measurements of entropy change proved difficult to learn, and we give the results from these experiments in Appendix B.

## 4   Experiments

In this section, we apply our EAMA approach to four different datasets across multiple domains. While the original motivation for this work came from healthcare applications, we also demonstrate on more general datasets and tasks that the EAMA approach can provide fast, high-quality solutions to the AMA problem. All experiments, including training the value models, are run on a dedicated compute node with 40 Intel Xeon Gold CPU cores and one V100 GPU with 32GB of VRAM, and we report training and inference times in this section. We first introduce the general structure for our experiments and define key metrics, then we investigate the EAMA algorithm for each specific application. Additional experiment details and ablation studies are given in appendices B and C.

**Experiment Setup.** Although we apply our EAMA approach to multiple real-world datasets, we do not have access to real, sample-specific modality acquisition costs. Therefore, we adopt a simple procedure to simulate heterogeneous acquisition costs. Specifically, for each sample $(\mathbf{X}_i, y_i)$, we sample a cost vector $\mathbf{z}_i' \sim \mathcal{N}(\mathbf{1}, \mathbf{I})$, then project it onto the positive orthant, so that $\mathbf{z}_i = \max(0, \mathbf{z}_i')$. For each dataset, we run multiple experiments, varying the cost magnitude $\lambda \in \mathbb{R}_+$ so that $\lambda \cdot \mathbf{z}_i' \sim \mathcal{N}(\lambda \cdot \mathbf{1}, \lambda^2 \cdot \mathbf{I})$. This simple parameterization allows us to evaluate each method with heterogeneous costs across different cost scales. Since we frame our value functions in terms of model accuracy, we construct a corresponding measure of reward based on the improvement in model accuracy by following a given policy. We first define the *baseline accuracy* as the accuracy of the classifier using the context of each sample only,

$$a_b = \frac{1}{n} \sum_{i=1}^n \mathbf{1}\{f_c(\mathbf{c}_i) = y_i\}. \tag{8}$$

For convenience, we define a modality acquisition policy $\pi : \mathcal{S} \times \mathbb{R}_+^M \to \mathcal{S}$ as any process that begins with the sample context modality $\mathbf{c}_i$ and acquisition costs $\mathbf{z}_i$ and returns the context along with a selected subset $p_i$ of modalities, $(\mathbf{c}_i, \mathbf{X}_i^{p_i})$, such as our EAMA method given in Algorithm 1. We then define the accuracy under $\pi$ as

$$a(\pi) = \frac{1}{n} \sum_{i=1}^n \mathbf{1}\{f_c(\pi(\mathbf{c}_i, \lambda \cdot \mathbf{z}_i)) = y_i\}. \tag{9}$$

Since we select subsets to maximize the expected change in accuracy minus acquisition costs, we therefore define our reward metric as

$$R(\pi) = (a(\pi) - a_b) - \frac{\lambda}{n} \sum_{i=1}^n \mathbf{z}_i^\top \mathbf{p}_i, \tag{10}$$

where $\mathbf{p}_i$ is the binary vector representation of the subset $p_i$ of acquired modalities. This reward metric captures a policy's ability to balance accuracy gain against acquisition costs. While the constant $a_b$ is not necessary when comparing policies, it shifts the reward such that $R(\pi) > 0$ indicates that the policy is able to effectively trade-off acquisition costs for accuracy gains. In addition to calculating the reward, for each dataset, we estimate the *model confusion rate*, $\mathbb{E}[Q_f^C]$, empirically. To do so, we randomly sample modality subsets $p, q \subset [M]$ following Algorithm 2 for the training dataset, and compute the empirical estimate $\mathbb{E}[Q_f^C] \approx \frac{1}{n} \sum_{i=1}^n Q_f^C(\mathbf{c}_i, \mathbf{X}_i^p, \mathbf{X}_i^q, y_i)$.

**Baseline Comparisons.** The final step in our experiment setup is establishing baseline policies to compare against our EAMA algorithm. Of the alternative methods outlined in Table 1, we identified the Efficient Modality Selection (EMS) algorithm (He et al., 2024) as the only scalable approach that could be reasonably evaluated with heterogeneous costs. Specifically, we use the greedy submodular maximization algorithm

(Nemhauser et al., 1978) to construct a sequence of modality subsets $s_1, \ldots, s_M$, adding one modality at each step, such that the ordering maximizes the accuracy gain at each step. The algorithm is summarized in Appendix C. We can define a sequence of static policies $\pi_1, \ldots, \pi_M$ such that $\pi_k$ corresponds to always selecting the first $k$ modalities from this greedy algorithm. Then, for each experiment, and each value of $\lambda$, we define $R_{static} = \max_{k \in [M]} R(\pi_k)$, the static greedy policy that achieves the best trade-off of accuracy gain against acquisition costs. Note that since the acquisition costs are i.i.d., greedily maximizing the expected reward is equivalent to maximizing the accuracy gain at each step. We provide a lower bound for the performance of the static EMS algorithm in Appendix C. This establishes a reasonable static baseline for us to measure the benefit of adaptive methods. To further isolate the benefits of our context-based EAMA approach, we extend the EMS method to adapt to the individual costs of each sample. Specifically, for any sample with acquisition costs, $\mathbf{z}_i$, the *adaptive EMS policy* selects

$$\pi_{\text{adapt}}(\lambda \cdot \mathbf{z_i}) = \underset{k \in [M]}{\arg\max}\, a(\pi_k) - \lambda \cdot \mathbf{z}_i^\top \mathbf{s}_k,$$

where $\mathbf{s}_k$ is the binary vector representation of the subset of modalities $s_k$ corresponding to the first $k$ modalities from the greedy EMS algorithm. Note that here, to simplify, we use abuse of notation, defining the adaptive policy as returning the index of the greedy modality subset rather than the state itself. This policy provides a principled, cost-adaptive baseline that scales well and allows us to further isolate the benefit of using information about the observed modalities and context in our EAMA algorithm compared to using population-level performance metrics and sample-level acquisition costs alone. Unlike the static EMS policy, the adaptive EMS policy can take advantage of samples with abnormally high or low acquisition costs, and adjusts the selected subset of modalities accordingly. In addition to the EMS baselines, we also experiment with multiple polynomial-time algorithms for optimizing the value model itself, and report the results of these ablation studies in Appendix B.

**CMU-MOSEI.** We apply our EAMA approach to the CMU-MOSEI dataset (Bagher Zadeh et al., 2018), a collection of $N = 23453$ video segments recording 1000 different speakers, annotated for sentiment. For our experiments we use the pipeline from the original paper to split the dataset 70/10/10 into train/validation/test datasets, and generate embeddings of the facial features of the speakers using (Stöckli et al., 2018) and the text generated from video transcription. We focus on binary sentiment classification for this experiment. To simulate the presence of many correlated modalities, we further partition the facial and text embeddings into smaller, disjoint embeddings. That is, we take the text embedding $\mathbf{X}^t \in \mathbb{R}^{300}$ and partition it into 10 chunks, $\mathbf{x}^{t,1}, \ldots, \mathbf{x}^{t,10} \in \mathbb{R}^{30}$, so that $\mathbf{x}^{t,1} \cup \cdots \cup \mathbf{x}^{t,10} = \mathbf{X}^t$. Similarly, we partition the facial feature embedding $X^v \in \mathbb{R}^{35}$ into three chunks, $\mathbf{x}^{v,1}, \mathbf{x}^{v,2} \in \mathbb{R}^{10}$ and $\mathbf{x}^{v,3} \in \mathbb{R}^{15}$, so that $\mathbf{x}^{v,1} \cup \mathbf{x}^{v,2} \cup \mathbf{x}^{v,3} = \mathbf{X}^v$. We take the first text chunk as the context, so that $\mathbf{c} = \mathbf{x}^{t,1}$. Then our final datset takes the form $(\mathbf{c}_i, \mathbf{x}^{t,2}, \ldots, \mathbf{x}^{t,10}, \mathbf{x}^{v,1}, \mathbf{x}^{v,2}, \mathbf{x}^{v,3})$ with target $y_i \in \{0, 1\}$, corresponding to a total of $M = 12$ observable modalities. We train a FFN for classification, using ModDrop (Liu et al., 2022) during training to ensure the network is robust to missing modalities. The value model was trained for 300 epochs, with an average wall time of 15 seconds per epoch.

Similar to the PTB-XL ECG and MIMIC-III experiments, we find that the EAMA algorithm with the accuracy change value function consistently outperforms the alternative methods. In this case, we estimate the model confusion rate to be $\mathbb{E}[Q_f^C] \approx 0.0481$, meaning that incorporating additional modalities confuses the classifier almost 5% of the time. Accordingly, we see a significant gap in performance between using the accuracy change and bit-flip value functions. For larger cost magnitudes, the average reward using the accuracy change value function is more than $4\times$ the average reward from using the bit-flip value function, which itself consistently outperforms the static and adaptive EMS policies. The EAMA algorithm achieves these results while solving the multi-stage optimization problem in 95 milliseconds per sample on average over the test dataset. For example, when $\lambda = 0.05$, the EMS static policy, optimized in this case by selecting only the top-1 most important modality to acquire, achieves an accuracy of 65.36% with an average cost of 0.054, while the EAMA algorithm with the accuracy change value function achieves an accuracy of 69.35% with an average cost of 0.014. The EAMA algorithm with the bit-flip value function achieves a greater accuracy of 72.07% but, being overly optimistic, incurs an average cost of 0.105, about $7.5\times$ more expensive. The adaptive EMS policy clearly improves upon the static EMS policy, but still clearly underperforms our

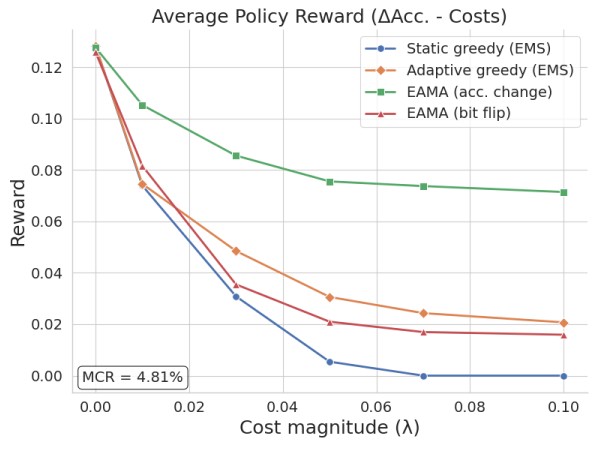
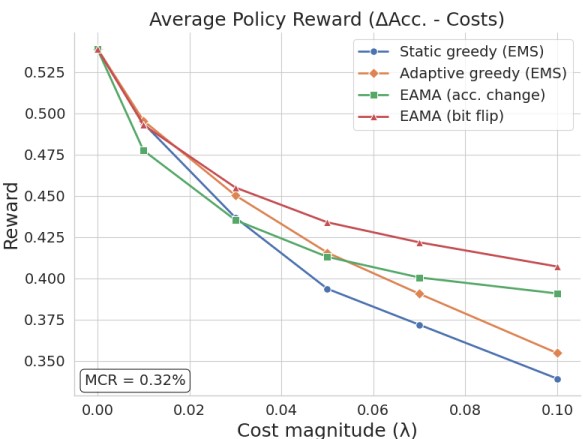

(a) Main results for the CMU-MOSEI dataset.  (b) Main results for the Patch-MNIST dataset.

Figure 4: We plot the average reward (Equation 10), defined as the accuracy gain from acquired modalities minus their acquisition costs, where costs are randomly sampled per input and scaled by $\lambda$. The bottom-left inset reports the empirical model confusion rate (MCR; probability that added modalities flip a correct prediction). When the MCR is non-negligible, as in the CMU-MOSEI experiments, EAMA with the accuracy-change value function performs best; when the MCR is near zero, as in Patch-MNIST, the bit-flip value function achieves the strongest results.

context-aware EAMA approach, in many cases returning less than $1/3$ the average reward of our method. The full results are given in Appendix B.

**Patch-MNIST.** We apply our EAMA approach to the MNIST dataset, a well-known dataset of grayscale hand-written digits of size $28 \times 28$ (LeCun et al., 2002). Similar to He et al. (2024), we modify the dataset by dividing each image into 16 patches of size $7 \times 7$, which we refer to as the Patch-MNIST dataset. Using one of the center patches as the context, we can construct a dataset of the form $(\mathbf{c}_i, \mathbf{x}_i^1, \ldots, \mathbf{x}_i^{15}) \in \mathbb{R}^{16 \times 4 \times 4}$, with the standard ten-digit classification target $y_i \in [10]$. We train a 2D CNN on this dataset, using ModDrop (Liu et al., 2022) during training to ensure the network is robust to missing modalities. The value model was trained for 100 epochs with an average wall time of 122 seconds per epoch. Our main results are summarized in Figure 4b. Most notably, we observe that unlike the three other experiments, in this case using the bit-flip value function consistently yields the best average reward. This result, we hypothesize, stems from the estimated model confusion rate, $\mathbb{E}[Q_f^C] \approx 0.0032$, being a fraction of a percent. As a result, $Q_f^A \approx Q_f^B$, and with so many modalities, the easier learning problem also leads to better performance. That is, learning $Q_f^A$ requires the value model to try to estimate the likelihood of the classifier becoming confused by the addition of new modalities, but this phenomenon is so exceedingly rare in this situation that the value model struggles to learn, yielding worse overall performance. This result supports our claim in Section 3.4 that, in the absence of model confusion, when more modalities almost universally leads to the same or better accuracy, using the bit-flip value function will lead to improved performance.

In Figure 5, we visualize real examples from our Patch-MNIST experiments. For each example, our EAMA algorithm selects patches that lead to the classifier making the correct prediction, while the patches selected by the EMS algorithm lead to an incorrect prediction. We can observe that the EAMA algorithm adapts the subset of patches selected based on the current observations, efficiently filling in the image. Importantly, as illustrated in the example given in the first row of Figure 5, by enabling subset selection at each stage, EAMA can acquire all necessary modalities in fewer stages than a process that acquires modalities one at a time. Further, as we see in Figure 11c, allowing our algorithm to select subsets of modalities matches or exceeds the average reward of using our value model to select and observe modalities one at a time. The EAMA algorithm achieves these results while solving the multi-stage optimization problem in 0.19 seconds per sample on average over the test dataset. These results illustrate how our EAMA algorithm can efficiently

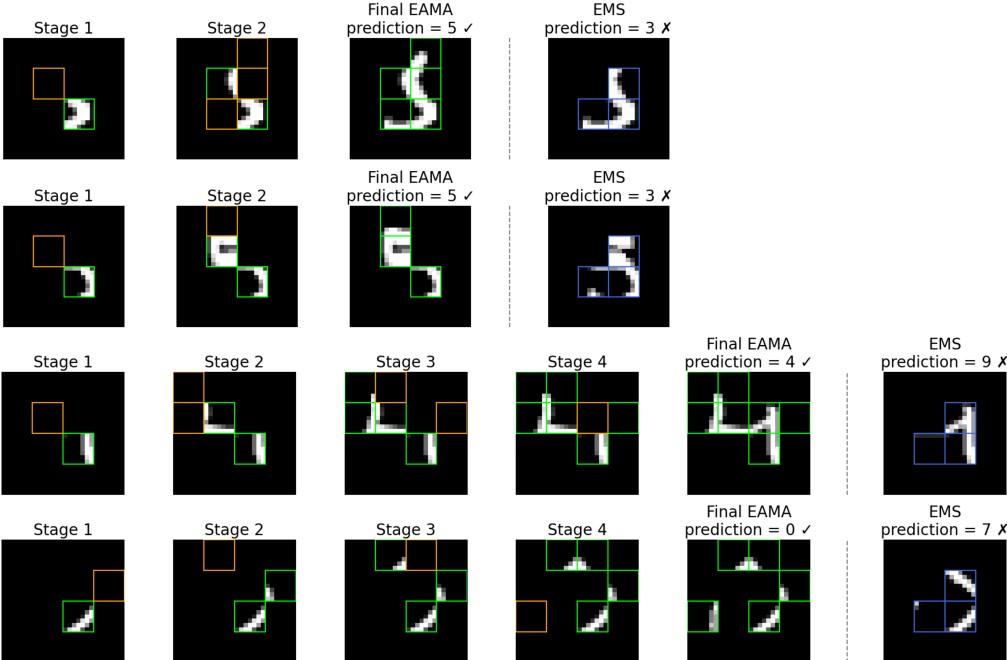

Figure 5: Examples of our EAMA algorithm applied to the Patch-MNIST dataset with cost magnitude $\lambda = 0.03$. At each stage, the patches in green have been observed, and the patches in orange are those selected by the EAMA algorithm to be acquired. On the right, we show the patches selected by the population-level EMS algorithm. In each of these cases, EAMA acquires patches that result in the classifier making the correct prediction, while the EMS algorithm selects patches that result in an incorrect prediction.

search a high-dimensional action space ($2^{15} = 32768$ subsets at stage one), balancing accuracy gains against heterogeneous acquisition costs, to find high-quality solutions to the AMA problem.

**MIMIC-III ED Diagnoses.** We apply our EAMA approach to the MIMIC-III dataset (Johnson et al., 2016), a large-scale collection of electronic health records (EHR) from patients admitted to critical care units at a large hospital in the United States. From this database, we select a cohort of patients who each received the following tests: an EKG, an echocardiogram, and a chest X-ray. We further select for patients who were diagnosed with one of the top-20 most common primary diagnoses, resulting in a final cohort of $N = 7373$ patients. The target for our classification problem is the primary diagnosis. We split the dataset temporally such that $N_{train} = 5769$, $N_{valid} = 739$, and $N_{test} = 865$. For each patient in our cohort, we extract the written notes for each of the three tests, in addition to a brief patient history, limited to 200 characters, which we use as the context. We extract separate embeddings for each of the notes using Clinical-Longformer (Li et al., 2022), a BERT-based encoder model for EHR data. Our final dataset has the form $(\mathbf{c}_i, \mathbf{x}_i^{ekg}, \mathbf{x}_i^{echo}, \mathbf{x}_i^{xray}) \in \mathbb{R}^{4 \times 768}$ with target $y_i \in [20]$, on which we train a FFN for classification, using ModDrop (Liu et al., 2022) during training to ensure the network is robust to missing modalities. The value model was trained for 100 epochs with an average wall time of 7 seconds per epoch.

We provide our main results for this dataset in Figure 7a, plotting the average reward on the test set for each approach over varying cost magnitudes. Note that unlike the other experiments, since $M = 3$ for this dataset, we find and report the optimal static policy over all subsets of modalities. We find empirically that $E[Q_f^C] \approx 0.01$ for this dataset, indicating that acquiring additional modalities "confuses" the classifier about 1% of the time. We observe that EAMA with the accuracy change value function consistently outperforms the static and adaptive EMS policies and EAMA with the bit-flip value function. This result holds even when $\lambda = 0$, corresponding to zero acquisition costs. Therefore, even in the absence of acquisition costs, we see evidence that the EAMA algorithm is able to anticipate the rare cases where additional modalities are likely to confuse the model, improving accuracy slightly over the optimal static policy, which in this case is to

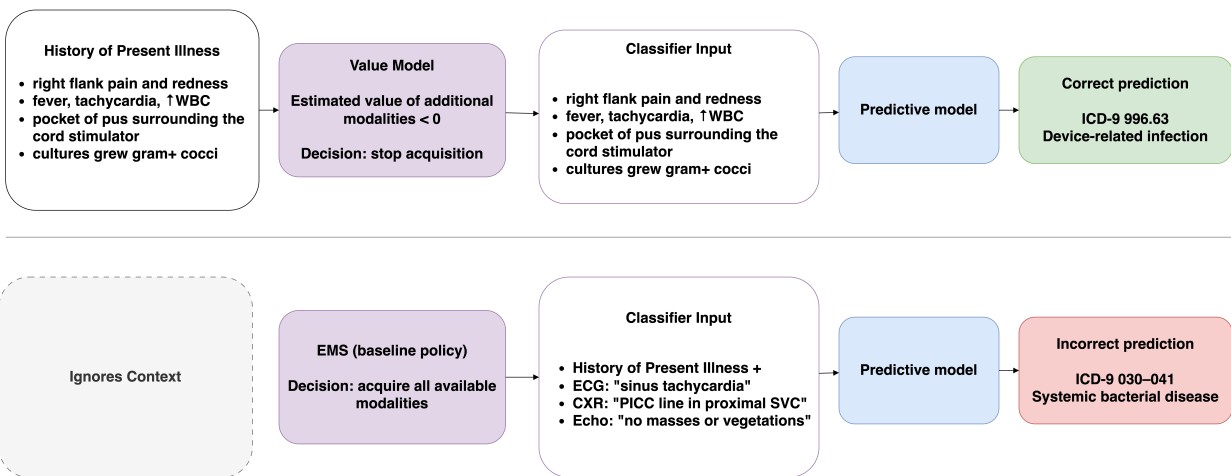

Figure 6: A real case from the MIMIC-III dataset, where a patient has developed a device-related infection. **Top Row:** using context from the patient's history of present illness, the EAMA value model recommends predicting a primary diagnosis without acquiring additional modalities. **Bottom Row:** acquiring all modalities, which is the best strategy on average across the population, confuses the classifier, leading it to make an incorrect primary diagnosis. This demonstrates a real example of model confusion.

use all of the modalities. In Figure 6, we illustrate a real example of model confusion from our experiments. In this case, our EAMA algorithm correctly recognizes that a primary diagnosis can be given based on the patient history alone. In contrast, by acquiring all the available tests, as recommended by the EMS policy, the classifier becomes confused, predicting a broader, non-specific diagnosis, which ultimately does not align with the actual primary diagnosis given to this patient. The EAMA algorithm achieves these results while solving the multi-stage optimization problem in 14 milliseconds per sample on average over the test dataset. We report our full experiment results in Appendix C.

**PTB-XL ECG Analysis.** We apply our EAMA approach to the PTB-XL ECG dataset, a collection of $N = 21799$ 12-lead clinical ECGs, each of 10 second duration, measured at 500Hz (Wagner et al., 2020). The target for this dataset is whether the ECG is normal or abnormal, a binary classification problem. We split the dataset using the standard folds provided, such that $N_{train} = 17418$, $N_{valid} = 2183$, and $N_{test} = 2198$. We use lead-I as our context, leaving the remaining $M = 11$ leads as observable modalities. Our dataset takes the form $(\mathbf{c}_i, \mathbf{X}_i) \in \mathbb{R}^{12 \times 5000}$ with label $y_i \in \{0, 1\}$. We train a 1D-CNN classifier on this dataset, using ModDrop (Liu et al., 2022) during training to ensure the network is robust to missing modalities. The value model was trained for 300 epochs with an average wall time of 84 seconds per epoch. The main results from this experiment are shown in Figure 7b. We observe again that the EAMA algorithm with the accuracy change value function consistently yields the best results in terms of average reward, across cost magnitudes. Similar to our MIMIC experiment, we again see that even in the absence of acquisition costs, our EAMA algorithm is able to identify when incorporating additional modalities is likely to confuse the classifier, improving the model accuracy over the static and adaptive EMS policies. Further, we note that with an estimated model confusion of $\mathbb{E}[Q_f^C] \approx 0.0366$, the bit-flip value function being unable to capture this model confusion phenomenon causes it to underperform relative to the accuracy change value function. This is largely due to our observation in Equation 7 that across multiple stages, the bit-flip value function will be overly optimistic in its estimate of accuracy gain. In this case, we see that when $\lambda = 0.01$, the bit-flip value function leads to EAMA acquiring 84.6% of the modalities on average, whereas the accuracy change value function leads EAMA to acquire 27.4% of the modalities on average. The EAMA algorithm achieves these results while solving the multi-stage optimization problem in 78 milliseconds per sample on average over the test dataset.

**EAMA Efficiently Solves AMA.** Across our experiments, we observe that the EAMA algorithm is able to find high-quality solutions to the AMA problem, solving the contextual, multi-stage subset selection

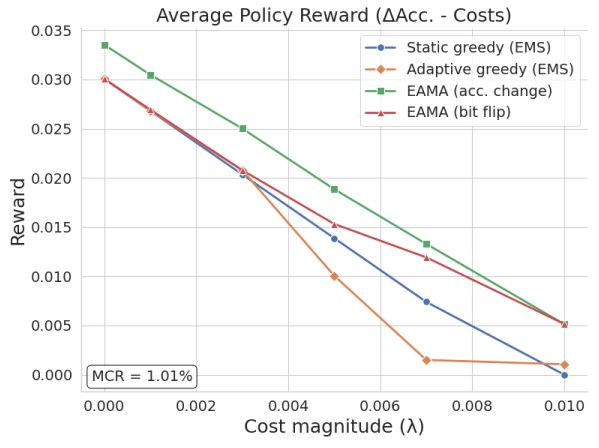 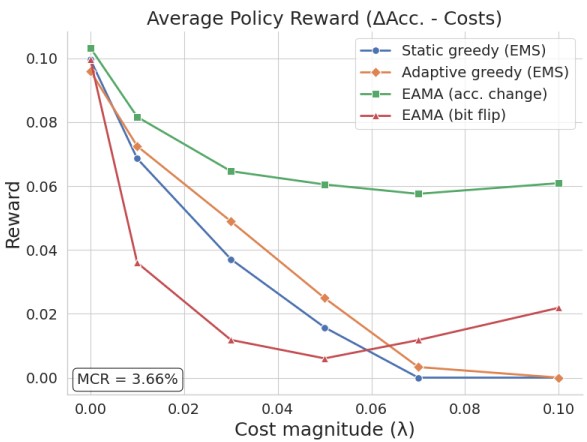

(a) Main results for the MIMIC-III dataset.  (b) Main results for the PTB-XL ECG dataset.

Figure 7: We plot the average reward (Equation 10), defined as the accuracy gain from acquired modalities minus their acquisition costs, where costs are randomly sampled per input and scaled by the parameter $\lambda$. The bottom-left inset reports the empirical model confusion rate (MCR), measuring the probability that adding modalities changes a correct prediction to an incorrect one. In these experiments, the MCR is non-negligible, and EAMA with the accuracy-change value function achieves the best performance.

problem with unobserved items in less than 14 milliseconds per instance when GPU-accelerated for $M = 3$, increasing up to only 0.2 seconds on average when $M = 15$. Taking the full EAMA pipeline into account, we find that we are able to train high-quality value models in at most 10 hours using a single V100 GPU with 32GB RAM, such that this time could potentially be reduced significantly with increased compute resources. Even in the zero-cost setting, EAMA is able to increase accuracy without using all modalities in some cases. With the appropriate value function, EAMA can achieve over an $8\times$ improvement in balancing accuracy gains against acquisition costs over static baselines, and a $6\times$ improvement over cost-adaptive baselines. Comparing against these baselines, we are able to isolate the benefits of our EAMA method, highlighting the importance of scalable, adaptive, and context-aware solutions to the AMA problem.

**MCR Influences Design and Performance.** Our experiments exhibit varying MCRs, and the empirical results support our theoretical insights regarding value function choice based on the MCR. In particular, we observe that for the Patch-MNIST dataset, where the MCR is near-negligible (about 0.32%), using the bit-flip value function clearly dominates the accuracy-gain value function, while the result is exactly the opposite for the PTB-XL ECG and CMU-MOSEI datasets, where the MCR was much more significant (3.66% and 4.81% respectively). Further, observing that for the MIMIC-III dataset, the two value functions perform similarly, especially in high-cost scenarios, with MCR = 1.01%, this suggests that if MCR < 1%, it is likely better to use the bit-flip value function, and otherwise, the accuracy change value function should be used. Note that the MCR also corresponds to how imbalanced the learning problem is for the accuracy change value function. The MCR being very small means the fraction of samples for which $Q_f^A = -1$ will be small, and the learning problem will be heavily imbalanced, which could impact the performance of the value model overall. These results reinforce the importance of selecting the correct value function, as well as the key role the MCR plays in understanding the AMA problem for any given application. In the absence of costs, the MCR most directly addresses the question of whether or not more is always better.

**Ablation Studies.** In Appendix B, we provide our full set of ablation studies, but we highlight one key ablation study here. We investigate the impact of the optimization algorithm used to maximize the objective $\phi(q)$ in Equation 3. In addition to the RUSM, Greedy, and Hybrid approaches outlined in Section 3.2, we also experiment with constraining the process to selecting and observing only one modality per stage, which we refer to as the Single-Item algorithm, increasing the number of stages while potentially decreasing risk. We visualize the results of this ablation study for the PTB-XL ECG and MIMIC-III datasets in Figure 8. Across experiments, we observe the same general trends when applying the various optimization algorithms. First,

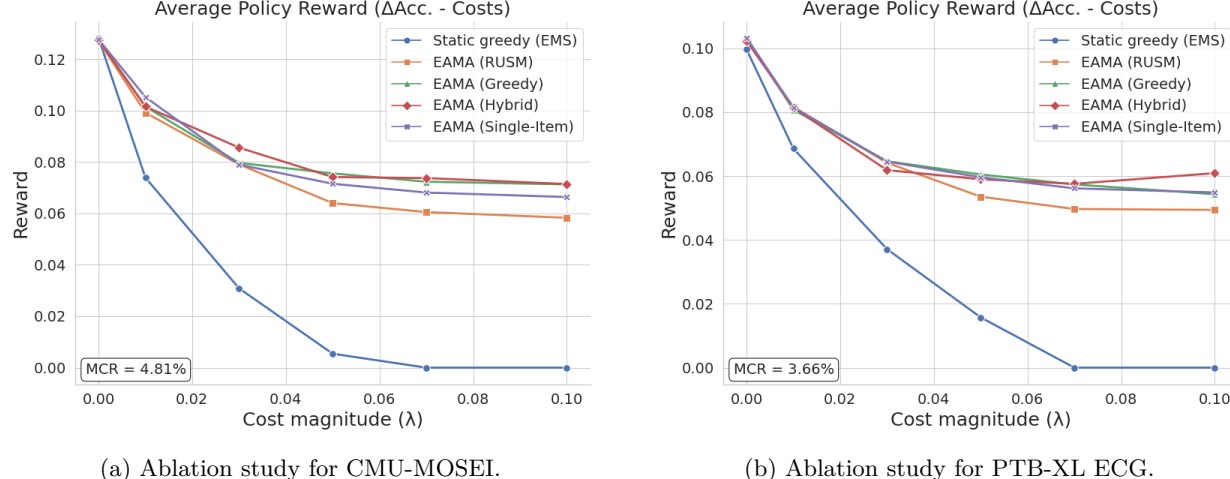

(a) Ablation study for CMU-MOSEI.

(b) Ablation study for PTB-XL ECG.

Figure 8: We plot the average reward (Equation 10), defined as the accuracy gain from acquired modalities minus their acquisition costs, where costs are randomly sampled per input and scaled by the parameter $\lambda$. The bottom-left inset reports the empirical model confusion rate (MCR), measuring the probability that adding modalities changes a correct prediction to an incorrect one. This ablation study shows the varying performance of different optimization algorithms for maximizing the objective in Equation 3.

the RUSM algorithm consistently underperforms the alternatives. The Greedy and Hybrid algorithms both perform well overall, with the Hybrid algorithm seeming to yield the most stable, high-quality solutions. The Hybrid algorithm often matches or exceeding the performance of the other methods, including the Single-Item algorithm. This result provides empirical evidence that we can solve the AMA problem, even for large $M$, with high enough quality that our EAMA algorithm can match or outperform the Single-Item approach, where one modality is acquired and observed at a time, suggesting that we can reduce the number of stages required to solve the AMA problem without sacrificing performance.

## 5 Conclusion

In this work, we provide a comprehensive and scalable methodology for solving the adaptive modality acquisition problem, which at its core is a challenging multi-stage subset selection problem with unobserved items and heterogeneous costs. We decompose this problem into two components: learning a value model that estimates the utility of acquiring any subset of modalities conditioned on the currently observed set, and optimizing modality selection at each stage to maximize value gain minus acquisition costs. For both components, we develop algorithms that scale linearly in the number of modalities, collectively referred to as EFFICIENT ADAPTIVE MODALITY ACQUISITION (EAMA). Central to our approach is the design and characterization of recursive value functions, learned as energy-based compatibility models that reason over arbitrary modality combinations in a sample-efficient and scalable manner. Our analysis yields a simple selection rule based on the model confusion rate, which determines the appropriate value function for a given task. Applied to a diverse set of real-world multimodal classification tasks with up to $M = 15$ modalities, EAMA achieves consistent improvements in balancing predictive performance against acquisition costs, including up to an $8\times$ improvement over static baseline methods and a $6\times$ improvement over adaptive baselines and, in some cases, improved accuracy while using fewer modalities. These results are achieved while inference for our EAMA method only requires 14 milliseconds per sample when $M = 3$ and 0.2 seconds when $M = 15$, scaling gracefully even as the number of modality subsets grows exponentially. Our framework provides a principled foundation for cost-aware, per-sample modality acquisition in real-world multimodal systems, supporting the deployment of efficient and adaptive multimodal models in domains such as healthcare, environmental monitoring, and scientific discovery.

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

## A    Properties of Value Functions

Here we provide proofs for the properties of the value functions defined in Section 3.

**Lemma 1.** *Given Assumption 3.1, the objective $\phi(q)$ is a non-monotonic submodular function.*

*Proof.* Expanding terms, we have

$$
\begin{aligned}
\phi(q_2 \cup \{j\}) - \phi(q_2) &= \widehat{Q}_f(\mathbf{c}, \mathbf{X}^p, q_2 \cup \{j\}) - \lambda \sum_{i \in q_2 \cup \{j\}} z_i - \widehat{Q}_f(\mathbf{c}, \mathbf{X}^p, q_2) + \lambda \sum_{i \in q_2} z_i \\
&= \widehat{Q}_f(\mathbf{c}, \mathbf{X}^p, q_2 \cup \{j\}) - \widehat{Q}_f(\mathbf{c}, \mathbf{X}^p, q_2) - \lambda z_j \\
&\leq \widehat{Q}_f(\mathbf{c}, \mathbf{X}^p, q_1 \cup \{j\}) - \widehat{Q}_f(\mathbf{c}, \mathbf{X}^p, q_1) - \lambda z_j \quad [\text{Assumption 3.1}] \\
&= \phi(q_1 \cup \{j\}) - \phi(q_1)
\end{aligned}
$$

Therefore, $\phi$ is a submodular function. Further, for any $q \subset [M]$, $j \in [M] \setminus q$ and $z_j > 0$, such that $\widehat{Q}_f(\mathbf{c}, \mathbf{X}^p, q \cup \{j\}) > \widehat{Q}_f(\mathbf{c}, \mathbf{X}^p, q)$, for all $\lambda > \frac{1}{z_j}(\widehat{Q}_f(\mathbf{c}, \mathbf{X}^p, q \cup \{j\}) - \widehat{Q}_f(\mathbf{c}, \mathbf{X}^p, q))$, we have that $\phi(q \cup \{j\}) < \phi(q)$, such that $\phi$ is a non-monotonic submodular function.  □

**Lemma 2.** *We define the bit flip value function as $Q_f^B(\mathbf{c}, \mathbf{X}^p, \mathbf{X}^q, y) = \mathbf{1}\{f_c(\mathbf{c}, \mathbf{X}^{p \cup q}) = y \ \& \ f_c(\mathbf{c}, \mathbf{X}^p) \neq y\}$ and the accuracy change value function as $Q_f^A(\mathbf{c}, \mathbf{X}^p, \mathbf{X}^q, y) = \mathbf{1}\{f_c(\mathbf{c}, \mathbf{X}^{p \cup q}) = y\} - \mathbf{1}\{f_c(\mathbf{c}, \mathbf{X}^p) = y\}$. If and only if $\forall p_1 \subseteq p_2 \subseteq [M]$, $f_c(\mathbf{c}, \mathbf{X}^{p_1}) = y \implies f_c(\mathbf{c}, \mathbf{X}^{p_2}) = y$, then $Q_f^B = Q_f^A$ and $\widehat{Q}_f^B = \widehat{Q}_f^A$.*

*Proof.* We can prove this claim by enumerating the cases directly. In the forward, direction, there are four cases. If $f_c(\mathbf{c}, \mathbf{X}^{p \cup q}) = y$ and $f_c(\mathbf{c}, \mathbf{X}^p) \neq y$, then $Q_f^P = Q_f^B = 1$. If $f_c(\mathbf{c}, \mathbf{X}^{p \cup q}) = y$ and $f_c(\mathbf{c}, \mathbf{X}^p) = y$, then

$Q_f^P = Q_f^B = 0$. Similarly, if $f_c(\mathbf{c}, \mathbf{X}^{p\cup q}) \neq y$ and $f_c(\mathbf{c}, \mathbf{X}^p) \neq y$, then $Q_f^P = Q_f^B = 0$. The fourth case, where $f_c(\mathbf{c}, \mathbf{X}^{p\cup q}) \neq y$ and $f_c(\mathbf{c}, \mathbf{X}^p) = y$, is impossible by our assumption, so we can conclude that $Q_f^B = Q_f^P$ everywhere. In the other direction, suppose $\exists p_1 \subseteq p_2 \subseteq [M]$, such that $f_c(\mathbf{c}, \mathbf{X}^{p_1}) = y$ and $f_c(\mathbf{c}, \mathbf{X}^{p_2}) \neq y$. Letting $q = p_2 \setminus p_1$, we see that $Q_f^P(\mathbf{c}, \mathbf{X}^{p_1}, \mathbf{X}^q, y) = -1$ and $Q_f^B(\mathbf{c}, \mathbf{X}^{p_1}, \mathbf{X}^q, y) = 0$, such that $Q_f^P \neq Q_f^B$ somewhere, concluding our proof.

We can also view the statement with respect to the expected value functions, $\widehat{Q}_f^B$ and $\widehat{Q}_f^P$. To relate the two functions, we observe

$$\begin{aligned}
\widehat{Q}_f^B(\mathbf{c}, \mathbf{X}^p, q) &= \mathbb{E}[\mathbf{1}\{f_c(\mathbf{c}, \mathbf{X}^{p\cup q}) = y \ \wedge \ f_c(\mathbf{c}, \mathbf{X}^p) \neq y\} \mid \mathbf{c}, \mathbf{X}^p] \\
&= P(f_c(\mathbf{c}, \mathbf{X}^{p\cup q}) = y \ \wedge \ f_c(\mathbf{c}, \mathbf{X}^p) \neq y \mid \mathbf{c}, \mathbf{X}^p) \\
&= \Big( P(f_c(\mathbf{c}, \mathbf{X}^{p\cup q}) = y \mid \mathbf{c}, \mathbf{X}^p) - P(f_c(\mathbf{c}, \mathbf{X}^p) = y \mid \mathbf{c}, \mathbf{X}^p) \Big) \\
&\quad + P(f_c(\mathbf{c}, \mathbf{X}^{p\cup q}) \neq y \ \wedge \ f_c(\mathbf{c}, \mathbf{X}^p) = y \mid \mathbf{c}, \mathbf{X}^p) \\
&= P(f_c(\mathbf{c}, \mathbf{X}^{p\cup q}) = y \mid \mathbf{c}, \mathbf{X}^p) - P(f_c(\mathbf{c}, \mathbf{X}^p) = y \mid \mathbf{c}, \mathbf{X}^p) \\
&\quad \big[ \iff \ f_c(\mathbf{c}, \mathbf{X}^p) = y \ \Rightarrow \ f_c(\mathbf{c}, \mathbf{X}^{p\cup q}) = y \big] \\
&= \widehat{Q}_f^P(\mathbf{c}, \mathbf{X}^p, q).
\end{aligned}$$

From this expansion, we can also observe that in the case when $Q_f^A$ is *not monotonic in q*, the difference between $\widehat{Q}_f^B$ and $\widehat{Q}_f^A$ is exactly $P(f_c(\mathbf{c}, \mathbf{X}^{p\cup q}) \neq y \wedge f_c(\mathbf{c}, \mathbf{X}^p) = y \mid \mathbf{c}, \mathbf{X}^p)$, the probability of the model becoming "confused" by the new information and flipping from a correct to an incorrect prediction. □

**Lemma 3.** *Let* $Q_f^A(\mathbf{c}, \mathbf{X}^p, \mathbf{X}^q, y) = \mathbf{1}\{f_c(\mathbf{c}, \mathbf{X}^{p\cup q}) = y\} - \mathbf{1}\{f_c(\mathbf{c}, \mathbf{X}^p) = y\}$*, so* $Q_f^A \in \{-1, 0, 1\}$*. Suppose a value model* $\widehat{Q}_f(\mathbf{c}, \mathbf{X}^p, q; \theta) \in \mathbb{R}^3$ *is trained with cross-entropy loss to predict* $Q_f^A$*, and define* $q_k(\mathbf{c}, \mathbf{X}^p, q) = $ softmax$(\widehat{Q}_f(\mathbf{c}, \mathbf{X}^p, q; \theta))_k$ *for* $k \in \{-1, 0, 1\}$*. At the population optimum,*

$$q_k(\mathbf{c}, \mathbf{X}^p, q) = P(Q_f^A = k \mid \mathbf{c}, \mathbf{X}^p, q),$$

*and therefore*

$$q_1(\mathbf{c}, \mathbf{X}^p, q) - q_{-1}(\mathbf{c}, \mathbf{X}^p, q) = \widehat{Q}_f^A(\mathbf{c}, \mathbf{X}^p, q),$$

*where*

$$\widehat{Q}_f^A(\mathbf{c}, \mathbf{X}^p, q) = \mathbb{E}_{y, \mathbf{X}^q}\left[Q_f^A(\mathbf{c}, \mathbf{X}^p, \mathbf{X}^q, y) \mid \mathbf{c}, \mathbf{X}^p, q\right].$$

*Proof.* Let $Z = (\mathbf{c}, \mathbf{X}^p, q)$ and define $p_k(Z) = P(Q_f^A = k \mid Z)$ for $k \in \{-1, 0, 1\}$. The conditional population cross-entropy risk of a probabilistic predictor $q(\cdot \mid Z)$ is

$$\mathcal{L}(q \mid Z) = - \sum_{k \in \{-1, 0, 1\}} p_k(Z) \log q_k(Z), \qquad \text{s.t.} \ \sum_k q_k(Z) = 1.$$

Using

$$\mathcal{L}(q \mid Z) = H(p(\cdot \mid Z)) + \mathrm{KL}(p(\cdot \mid Z) \| q(\cdot \mid Z)),$$

the minimum is attained when $q_k(Z) = p_k(Z)$ for all $k$. Since $Q_f^A \in \{-1, 0, 1\}$,

$$\mathbb{E}[Q_f^A \mid Z] = p_1(Z) - p_{-1}(Z) = q_1(Z) - q_{-1}(Z).$$

Taking the conditional expectation over $y$ and $\mathbf{X}^q$ given $(\mathbf{c}, \mathbf{X}^p, q)$ yields

$$\mathbb{E}[Q_f^A \mid \mathbf{c}, \mathbf{X}^p, q] = \mathbb{E}_{y, \mathbf{X}^q}\left[Q_f^A(\mathbf{c}, \mathbf{X}^p, \mathbf{X}^q, y) \mid \mathbf{c}, \mathbf{X}^p, q\right] = \widehat{Q}_f^A(\mathbf{c}, \mathbf{X}^p, q),$$

which completes the proof. □

# B   Ablation Studies

**Optimization Algorithms for Reward Maximization.** In Section 3 we introduce the Randomized USM (RUSM) algorithm (Buchbinder et al., 2012) for maximizing an unconstrained, non-monotonic submodular function as a method for optimizing the objective $\phi(q)$ in Equation 3. Under Assumption 3.1, the RUSM algorithm gives a tight 1/2-approximation guarantee for maximizing Equation 3, with the run-time scaling in $O(M)$. However, this algorithm is primarily designed to mitigate the worst-case scenarios, which can occur when adding an element to the set early in the process can become extremely costly later on. This necessitates the second set, $b$, in Algorithm 1, which is initialized as $b = [M]$ to consider the cost of removing elements from the larger set. In the case of modality acquisition, such harmful interaction effects between modalities are unlikely, and most of the non-monotonicity of $\phi$ is driven by the acquisition costs. In this sense, the RUSM algorithm can be overly pessimistic, potentially removing modalities early in the process because the benefit of saving costs by removing the element from $b$ outweighs the marginal gain of adding it to the set $a$.

An alternative to the RUSM algorithm is the standard greedy submodular maximization algorithm (Nemhauser et al., 1978), which iteratively adds modalities, maximizing marginal gain in the objective value. The run-time for this algorithm scales in $O(M^2)$ and, under the assumption that $\phi$ is monotonic, enjoys a $(1 - e^{-1})$-approximation guarantee. Note that if $\phi$ is non-monotonic, in the worst case, this greedy algorithm can yield arbitrarily poor solutions. Unlike the RUSM algorithm, this greedy algorithm is optimistic, and will not remove modalities from consideration early in the process. Therefore, in addition to the RUSM algorithm, we implement and evaluate the greedy algorithm (Greedy) for each experiment, with the results shown in Figure 11. In addition to RUSM and Greedy, we combine the two approaches, applying both algorithms and taking the whichever solution achieves a greater objective value. In Figure 11, we refer to this as the Hybrid algorithm. This hybrid approach still scales in $O(M^2)$ and enjoys the 1/2-approximation guarantee of the RUSM algorithm, while incorporating the potential for the more optimistic Greedy algorithm to find better solutions that RUSM might miss. Finally, for comparison, we restrict the problem to finding the single modality that maximizes the marginal reward gain at each stage, rather than selecting a subset of modalities. This approach, which we refer to as Single-Item in Figure 11, demonstrates the reward when we have the flexibility to acquire and observe modalities one at a time. In practice this approach is often unfeasible, as it is often best to reduce the total number of stages before making a prediction.

Across experiments, we observe the same general trends when applying the various optimization algorithms. First, the RUSM algorithm consistently underperforms the alternatives, likely due to being overly conservative and removing too many modalities. The Greedy and Hybrid algorithms both perform well overall, with the Hybrid algorithm seeming to yield the most stable, high-quality solutions. The Hybrid algorithm often matches or exceeding the performance of the other methods, including the Single-Item algorithm. This result provides empirical evidence that we can solve the subset selection problem with unobserved items, even for large $M$, with high enough quality that our EAMA algorithm can match or outperform the Single-Item approach, where one modality is acquired and observed at a time. This result suggests that we can significantly reduce the number of stages required to solve the AMA problem without sacrificing performance.

**Impact of State on Value Models.** In our experiments, the value models estimate the expected value of acquiring a subset of modalities, taking the currently observed modalities as input. In addition to the state $(\mathbf{c}, \mathbf{X}^p)$, where $p \subset [M]$ is the set of observed modalities, we encode the set $p$ itself and pass it as an input to the value model, in addition to the encoded subset $q \subseteq [M] \setminus p$ of proposed modalities to acquire. To measure the importance of using the state in value estimation, for each dataset, we train a value model both with and without the state as an input to the model. That is, for one value model, the input is of the form $(\mathbf{c}, \mathbf{X}^p, p, q)$, and for the other value model, the input is $(p, q)$ only. The latter value model can only learn the expected value of acquiring subset $q$, given subset $p$ has already been acquired, over all possible states. In Figure 13 we show the results of this ablation study, plotting the ROC-AUC score of the value model after each training epoch on a held-out validation dataset. Note that we fix a random seed when sampling the subsets $p, q \subset [M]$ for the validation dataset, so that the samples are deterministic and the ROC-AUC is comparable across epochs. We show these curves for the best value function for each dataset respectively. For the accuracy change value function, we report the one-versus-one ROC-AUC score. We

---

**Algorithm 3** RandomizedUSM for AMA (Buchbinder et al., 2012))

---

**Require:** Context modality $\mathbf{c}$, sample view $\mathbf{X}^p$ expected value function $\widehat{Q}_f$, cost vector $\mathbf{z}$, cost parameter $\lambda$

    $U \leftarrow [M] \setminus p$                                                        $\triangleright$ Unobserved modalities

    $a \leftarrow \emptyset,\ b \leftarrow U$

    **for** $j \in U$ **do**

        $a_j \leftarrow \phi(a \cup \{j\};\ \mathbf{c}, \mathbf{X}^p, \mathbf{z}, \lambda) - \phi(a;\ \mathbf{c}, \mathbf{X}^p, \mathbf{z}, \lambda)$

        $b_j \leftarrow \phi(b \setminus \{j\};\ \mathbf{c}, \mathbf{X}^p, \mathbf{z}, \lambda) - \phi(b;\ \mathbf{c}, \mathbf{X}^p, \mathbf{z}, \lambda)$

        $a_j' \leftarrow \max(a_j, 0), b_j' \leftarrow \max(b_j, 0)$

        **if** $a_j' + b_j' > 0$ **then**

            $p_j \leftarrow a_j'/(a_j' + b_j')$

        **else**

            $p_j \leftarrow 0$

        **end if**

        $z_j \sim \text{Bernoulli}(p_j)$

        **if** $z_j = 1$ **then**

            $a \leftarrow a \cup \{j\}$

        **else**

            $b \leftarrow b \setminus \{j\}$

        **end if**

    **end for**

    **Return** subset of modalities $a$ (equivalently $b$)

---

can observe that across experiments, the addition of the state as an input to the value model has a clear and significant impact on the performance of the value model, improving the ROC-AUC by at least 5% in absolute points, and up to 17% for the CMU-MOSEI dataset. This result indicates that, across datasets and tasks, the current state, including all observed modalities, plays an important role in determining the value of acquiring any of the remaining unobserved modalities.

**Information Gain Value Function.** In addition to the value functions defined in Section 3.4, we also explored a more granular value function, which measures the value in terms of *information gain*, defined as

$$Q_f^I(\mathbf{c}, \mathbf{X}^p, \mathbf{X}^q, y) = \log(f_s(c, \mathbf{X}^{p \cup q})_y / f_s(c, \mathbf{X}^p)_y), \tag{11}$$

the change in the entropy of the classifier's predictions by incorporating modalities $q$ given the context and acquired modalities. From a theoretical perspective, this value function has multiple desirable properties, as $Q_f^I$ is continuous, strongly convex, and telescoping. This was the first value function considered, as it provides the most granular measure of value gain. However, we found that the measurements of this value function are both noisy and high-variance, making prediction difficult. In Figure 12, we plot the MSE of the information gain value model on the validation dataset across each of our experiments. We use the same setup and value model architecture as in our experiments with the other value functions, only changing the output layer for regression and training with MSE-loss. In addition, we plot the baseline MSE on the validation dataset when we perform mean-value prediction. That is, we estimate the mean information gain value on the training dataset, $\hat{y} = \frac{1}{n} \sum_{i=1}^{n} Q_f^I(\mathbf{c}_i, \mathbf{X}_i^p, \mathbf{X}_i^q, y_i)$, and calculate the MSE on the test dataset when we predict $\hat{y}$ as a constant. It can be easily seen that, across experiments, the value models are unable to outperform mean-value prediction, indicating that the models cannot reliably learn from the state or the modality subsets in question to determine the change in cross-entropy loss. This result holds across experiments with a varying number of modalities, dataset structure, and model confusion rates. While it might be possible to improve upon these value models, these results suggest using other value functions will likely lead to better performance.

**Value Model Calibration.** As part of our experiments, we investigated the potential benefits of applying model calibration to the learned value models to better align their outputs with the true probability of each outcome (e.g. the classifier flipping from an incorrect to a correct prediction). Since we use the outputs of the value model directly in our EAMA algorithm to find the best subset given the costs, the outputs

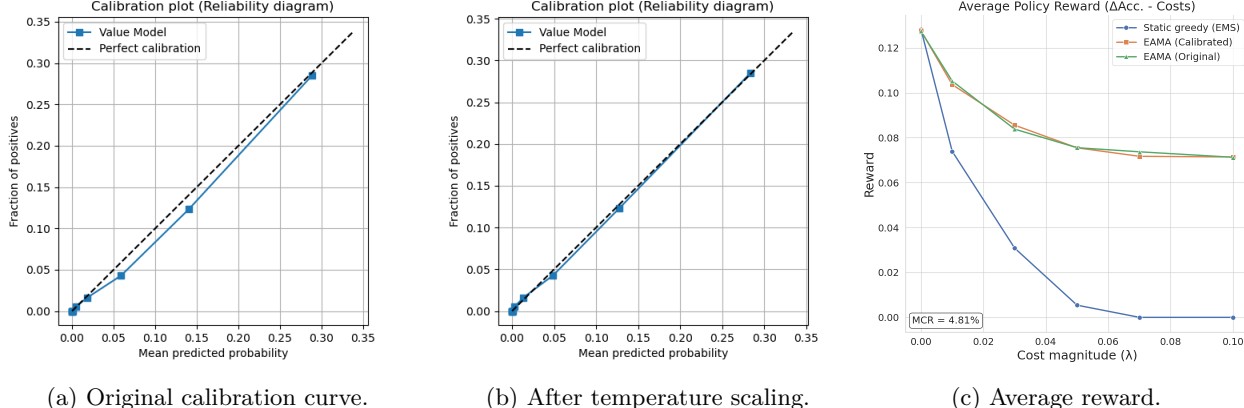

(a) Original calibration curve.    (b) After temperature scaling.    (c) Average reward.

Figure 9: Left: The calibration curve for the positive class of the original CMU-MOSEI value model. Middle: The calibration curve for the CMU-MOSEI value model after temperature scaling. Right: The average reward of the original and calibrated value models. Although temperature scaling improves value model calibration, we observe the impact on downstream performance to be negligible for the AMA problem.

should be well-calibrated so the cost-accuracy trade-off is properly aligned. To adjust the calibration of the learned value models, we apply temperature scaling to the softmax output, fitting the temperature parameter on the validation dataset (Guo et al., 2017). We visualize the calibration results for CMU-MOSEI in Figure 9. For the calibration plots, since we use the accuracy change value function for CMU-MOSEI, we plot the calibration of the positive class, corresponding to the probability of the classifier making a correct prediction after acquiring the proposed modalities. Across experiments, we found that while temperature scaling improved model calibration, the impact on downstream performance was negligible, as we can see in Figure 9. It is possible that if a value model is very poorly calibrated, this model calibration could be beneficial. For our experiments, we find this extra calibration step to be unnecessary for solving the AMA problem.

## C   Additional Experiment Details

**Validation of Submodularity Assumption.** Core to the methodological development in this work is the underlying assumption that the value of adding modalities to a multimodal learning system is submodular, exhibiting diminishing marginal returns. Therefore, we investigate this assumption for the experiments presented in this work. Specifically, in Algorithm 1, we rely on the submodularity of the learned value model $\widehat{Q}_f(\cdot; \theta)$ in terms of the proposed subset of modalities to acquire. To validate our approach, we test the *second-order submodularity condtion* for each value model. In our setting, this condition can be stated as follows: $\widehat{Q}_f(\cdot; \theta)$ is submodular if and only if for any context modality $\mathbf{c}$, subset of acquired modalities $p \subset [M]$, subset of proposed modalities $q \subset [M] \setminus p$, and modalities $i, j \in [M] \setminus p \cup q$, we have

$$\Delta_{ij}(\mathbf{c}, \mathbf{X}, q, p) = \widehat{Q}_f(\mathbf{c}, \mathbf{X}^p, q \cup \{i, j\}; \theta) - \widehat{Q}_f(\mathbf{c}, \mathbf{X}^p, q \cup \{i\}; \theta) - \widehat{Q}_f(\mathbf{c}, \mathbf{X}^p, q \cup \{j\}; \theta) + \widehat{Q}_f(\mathbf{c}, \mathbf{X}^p, q; \theta) \leq 0.$$

We can define the pair-wise second-order condition as $\Delta_{ij} = \mathbb{E}_{\mathbf{c}, \mathbf{X}, q, p}[\Delta_{ij}(\mathbf{c}, \mathbf{X}, q, p)]$, which we compute empirically over the samples in the test set, randomly sampling subsets $q, p$ uniformly as in Algorithm 2. We then compute the average condition value over all pairs, $\bar{\Delta} = \mathbb{E}_{i,j}[\Delta_{i,j}]$, along with the 95%-confidence intervals for this metric. To further investigate rate and significance of the violations of this condition, we define the positive violation rate, $\text{PVR}_{1\sigma}$, as the fraction of examples where $\Delta_{i,j}(\mathbf{c}, \mathbf{X}, q, p) > 0$ and greater than one standard deviation above the mean value. To test the significance of these violations, for each modality pair $i, j$, we perform a one-sided t-test to test whether $\Delta_{i,j} > 0$, applying Benjamini-Hochberg multiple testing correction since we perform one test for each modality pair (Benjamini & Hochberg, 1995). We report the significant positive violation rate, $\text{SPVR}_{0.05}$ at a false discovery rate of 0.05. Our results are summarized in Table 2.

| Task | $\bar{\Delta}$ | $\bar{\Delta}\,[\ell_{0.95}, u_{0.95}]$ | $\mathrm{PVR}_{1\sigma}(\%\downarrow)$ | $\mathrm{SPVR}_{0.05}(\%\downarrow)$ |
|---|---|---|---|---|
| PTB-XL EKG | -0.0026 | [-0.0032, -0.0022] | 0.59 | 0.0 |
| Patch-MNIST | -0.0037 | [-0.0041, -0.0032] | 0.35 | 0.0 |
| MIMIC-III | -0.0313 | [-0.0335, -0.0290] | 4.5 | 0.0 |
| CMU-MOSEI | -0.0036 | [-0.0044, -0.0029] | 2.5 | 0.0 |

Table 2: Second-order submodularity condition for the final value models across each task. The condition $\bar{\Delta} < 0$ indicates that, in expectation, submodularity holds for the given function. Taking the 95%-CI upper bound for $\bar{\Delta}$, we see that the submodularity assumption safely holds across all experiments. There are no statistically significant positive violations of the second-order condition ($\mathrm{SPVR}_{0.05}$) across experiments.

Across our experiments, we observe that the mean and 95% confidence intervals for the second-order equation are negative, such that in expectation, the value models all behave as submodular functions. Importantly, there were zero significant positive violations found for any modality pair across all experiments. Although there were some positive violations for specific instances, as we can see from the $\mathrm{PVR}_{1\sigma}$ rate, none of these represented a significant pattern of positive violations for any pair of modalities. For example, for the MIMIC-III experiment, despite the 4.5% $\mathrm{PVR}_{1\sigma}$ rate, the test statistics for each pair of modalities was at most $t = -16.7$, indicating substantial confidence that the positive violations are not significant. Taken together, these results provide strong empirical evidence that the learned value models exhibit submodular structure across the domains considered, and that this assumption holds in our experiments.

**Efficient Modality Selection.** In Algorithm 4, we summarize the greedy submodular maximization algorithm of Nemhauser et al. (1978) in the context of modality selection, as introduced and characterized by He et al. (2024). We modify the objective function to match our cost-adjusted setting, iteratively adding modalities that achieve the maximum marginal gain in expected reward, as defined in Equation 10. For brevity, we denote by $\pi(p)$ the static policy which always acquires the subset $p \subseteq [M]$ of modalities. The accuracy function $a(\cdot)$, as defined in Equation 9, represents the out-of-sample accuracy for the classifier $f$ under some modality acquisition policy $\pi$. Scaling in $O(M^2)$, this greedy policy establishes a strong and scalable baseline algorithm. Note that in our experiment setup, we sample the acquisition cost $z_j$ i.i.d., such that maximizing the marginal reward in Algorithm 4 is equivalent to maximizing the marginal accuracy gain. Further, under the assumption that the accuracy function $a(\pi(p))$ is submodular in $p$, we can bound $a(\pi_k) \geq (1 - e^{-1})a(\pi_k^*)$, where $\pi_k^*$ is the optimal accuracy maximizing subset of size $k$. Finally, under i.i.d. acquisition costs, we can obtain the bound

$$\max_{k \in [M+1]} (a(\pi_k) - a_b) - \lambda \sum_{j \in p_k} \mathbb{E}[z_j] \geq (1 - e^{-1}) \cdot \mathrm{OPT}\left(\frac{\lambda}{1 - e^{-1}}\right), \tag{12}$$

where $\mathrm{OPT}(\lambda)$ corresponds to the optimal reward under a static modality acquisition policy with cost magnitude $\lambda$. Therefore, in our experiments, Algorithm 4 is provably competitive against the optimum of a slightly more regularized version of the problem. The cost-adjusted EMS algorithm therefore serves as a scalable and competitive baseline for solving the AMA problem.

**Learning to Encode Modality Subset Size.** In this work, to encode subsets of modalities, we use a simple mean of the independent modality embeddings, which are on the unit ball. We use the same modality embeddings to encode both observed and proposed modalities. This provides a simple, efficient encoding scheme that enables the value model to reason over an exponential number of possible modality subsets. However, mean pooling does not inherently encode the size of the subset, especially if the modalities are strongly positively correlated. Specifically, for embeddings $\mathbf{e}_1, \ldots, \mathbf{e}_M \in \mathbb{R}^d$ such that $||\mathbf{e}_i||_2 = 1$, if $\mathbf{e}_i = \mathbf{e}_j$ for all $i, j \in [M]$, then for any subset $p \subseteq [M]$, we have $||\mathbf{e}_p||_2 = 1$. Losing this information could negatively impact how well the value model can reason about the acquired and proposed modality subsets. To investigate this, we plot the average modality subset embedding magnitude across different subset sizes for each experiment, and measure the average cosine similarity between the modality embeddings. Note that $||\mathbf{e}_p||_2^2 = \frac{1}{|p|} + \frac{1}{|p|^2} \sum_{i \neq j} \mathbf{e}_i^\top \mathbf{e}_j$, such that if the embeddings are orthogonal, $\mathbf{e}_i^\top \mathbf{e}_j = 0$, and $||\mathbf{e}_p||_2 = \frac{1}{\sqrt{|p|}}$, so

---

**Algorithm 4** Cost-Adjusted Efficient Modality Selection ( (He et al., 2024; Nemhauser et al., 1978))

---

**Require:** Modalities $M$, multimodal classifier $f$, random cost vector $\mathbf{z}$, cost parameter $\lambda$

$p_0 \leftarrow \emptyset$

$\pi_0 \leftarrow \pi(p_0)$

$a_b = a(\pi_0)$        $\triangleright$ Baseline accuracy (context modality only).

**for** $i \in [M]$ **do**

    $j \leftarrow \arg\max_{j \in [M] \setminus p_i} (a(\pi(p_i \cup \{j\})) - a_b) - \lambda \mathbb{E}[z_j]$      $\triangleright$ Maximize marginal reward.

    $\pi_i \leftarrow \pi(p_i \cup \{j\})$

    $p_{i+1} \leftarrow p_i \cup \{j\}$

**end for**

$\pi_k = \arg\max_{k \in [M+1]} (a(\pi_k) - a_b) - \lambda \sum_{j \in p_k} \mathbb{E}[z_j]$

**Return** Greedy-optimal policy $\pi_k$

---

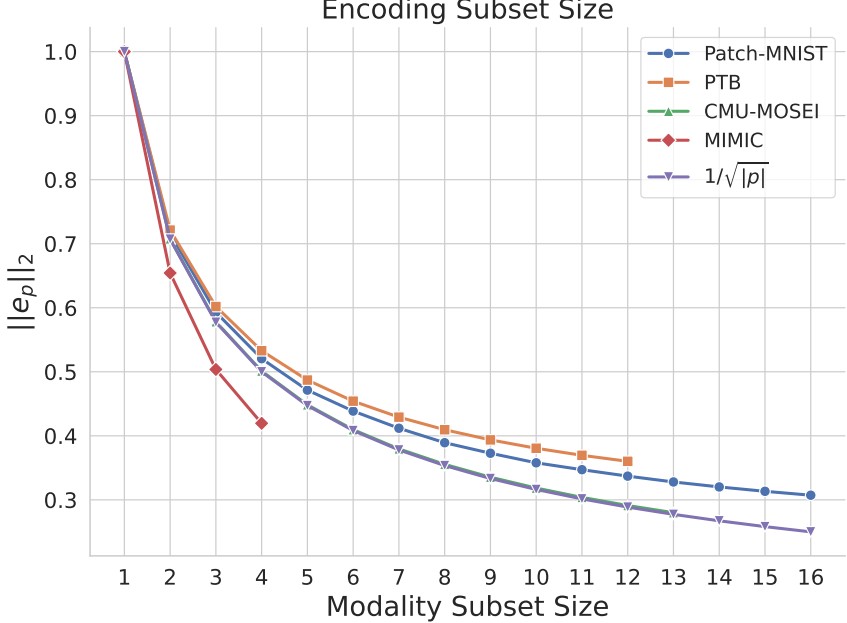

Figure 10: Comparing the magnitude of the modality subset mean embedding against subset size. If the embeddings were orthogonal, we would expect $||\mathbf{e}_p||_2 = 1/\sqrt{|p|}$. We observe that the learned embeddings roughly follow this curve, in some cases decreasing at a faster rate, such that subset size can be approximately inferred from the mean embedding alone.

the subset embedding magnitude is monotonic decreasing in the subset size. We plot the results in Figure 10. We observe that the learned embeddings roughly follow the expected curve if they were orthogonal, and all curves are monotonic decreasing. In the case of MIMIC-III, many of the learned embeddings have negative cosine similarity, such that they cancel under averaging and may encode subset cardinality even more strongly than approximately orthogonal embeddings would. For MIMIC-III, we have $\mathbb{E}[\mathbf{e}_i^\top \mathbf{e}_j] = -0.09$, for PTB-XL EKG we have $\mathbb{E}[\mathbf{e}_i^\top \mathbf{e}_j] = 0.05$, for CMU-MOSEI we have $\mathbb{E}[\mathbf{e}_i^\top \mathbf{e}_j] = 0.0015$, and for Patch-MNIST we have $\mathbb{E}[\mathbf{e}_i^\top \mathbf{e}_j] = 0.03$, such that no modality embeddings exhibit strong positive alignment. Therefore, even from a simple strategy like mean embedding, across experiments, the learned embeddings are sufficiently diverse such that properties like modality subset size can be approximately recovered. We note that future work could explore more expressive forms of set encoding, such as attention-based multiple instance learning (Ilse et al., 2018), that can adapt the representation of each modality in the subset based on the other modalities in the set.

**PTB-XL ECG Dataset.** For the PTB-XL ECG dataset, we trained a 1D-CNN with two convolutional layers per lead and a simple MLP classifier head with hidden size $[64, 32]$ and modality embedding dimension $d = 1024$. The input to the MLP head is the average embedding of the observed leads $x \in \mathbb{R}^{64}$ and the average embedding of the observed modalities. We train this model using the Adam optimizer (Kingma & Ba, 2014) with a learning rate of $1e - 4$ and batch size of 128 for 10 epochs. The value model shares the same architecture, except the MLP head takes as input the average lead embedding, the average embedding of the observed modalities, and the average embedding of the subset of modalities selected to acquire. The value model is trained for 300 epochs, also with a learning rate of $1e - 4$. and a batch size of 128. The full results for this experiment are given in Table 3.

**Patch-MNIST Dataset.** For the Patch-MNIST dataset, we trained a 2D-CNN with two convolutional layers and a simple MLP classifier head with hidden size $[128, 64, 32]$ and modality embedding dimension $d = 1024$. The input to the MLP head is the flattened feature vector $x \in \mathbb{R}^{9216}$ after processing the image through the convolutional layers, along with and the average embedding of the observed modalities. We train this model using the Adam optimizer (Kingma & Ba, 2014) with a learning rate of $1e - 4$ and batch size of 128 for 15 epochs. The value model shares the same architecture, except the MLP head takes as input the average lead embedding, the average embedding of the observed modalities, and the average embedding of the subset of modalities selected to acquire. The value model is trained for 100 epochs, also with a learning rate of $1e - 4$. and a batch size of 128. The full results for this experiment are given in Table 4.

**MIMIC-III Dataset.** For the MIMIC-III dataset, we first extracted embeddings $x \in \mathbb{R}^{768}$ for each of the four modalities using a Clinical-Longformer model, finetuned on our diagnostic prediction task. Concatenating the embeddings, the final input to our classifier are samples of shape $x \in \mathbb{R}^{3072}$. We trained a standard MLP with hidden size $[1028, 256]$ and modality embedding dimension $d = 16$. The input to the MLP head is the average embedding of the observed leads and the average embedding of the observed modalities. We train this model using the Adam optimizer (Kingma & Ba, 2014) with a learning rate of $1e-4$ and batch size of 256 for 10 epochs. The value model shares the same architecture, except the MLP head takes as input the average lead embedding, the average embedding of the observed modalities, and the average embedding of the subset of modalities selected to acquire. The value model is trained for 100 epochs, also with a learning rate of $1e - 4$. and a batch size of 128. The full results for this experiment are given in Table 5.

**CMU-MOSEI Dataset.** For the CMU-MOSEI dataset, we trained a standard MLP classifier with hidden size $[64, 32]$ and modality embedding dimension $d = 1024$. The input to the MLP is the feature vector $x \in \mathbb{R}^{335}$ of all the modalities concatenated together along with the average embedding of the observed modalities. We train this model using the Adam optimizer (Kingma & Ba, 2014) with a learning rate of $1e - 4$ and batch size of 128 for 10 epochs. The value model shares the same architecture, except the MLP head takes as input the average lead embedding, the average embedding of the observed modalities, and the average embedding of the subset of modalities selected to acquire. The value model is trained for 100 epochs, also with a learning rate of $1e - 4$. and a batch size of 128. The full results for this experiment are given in Table 6.

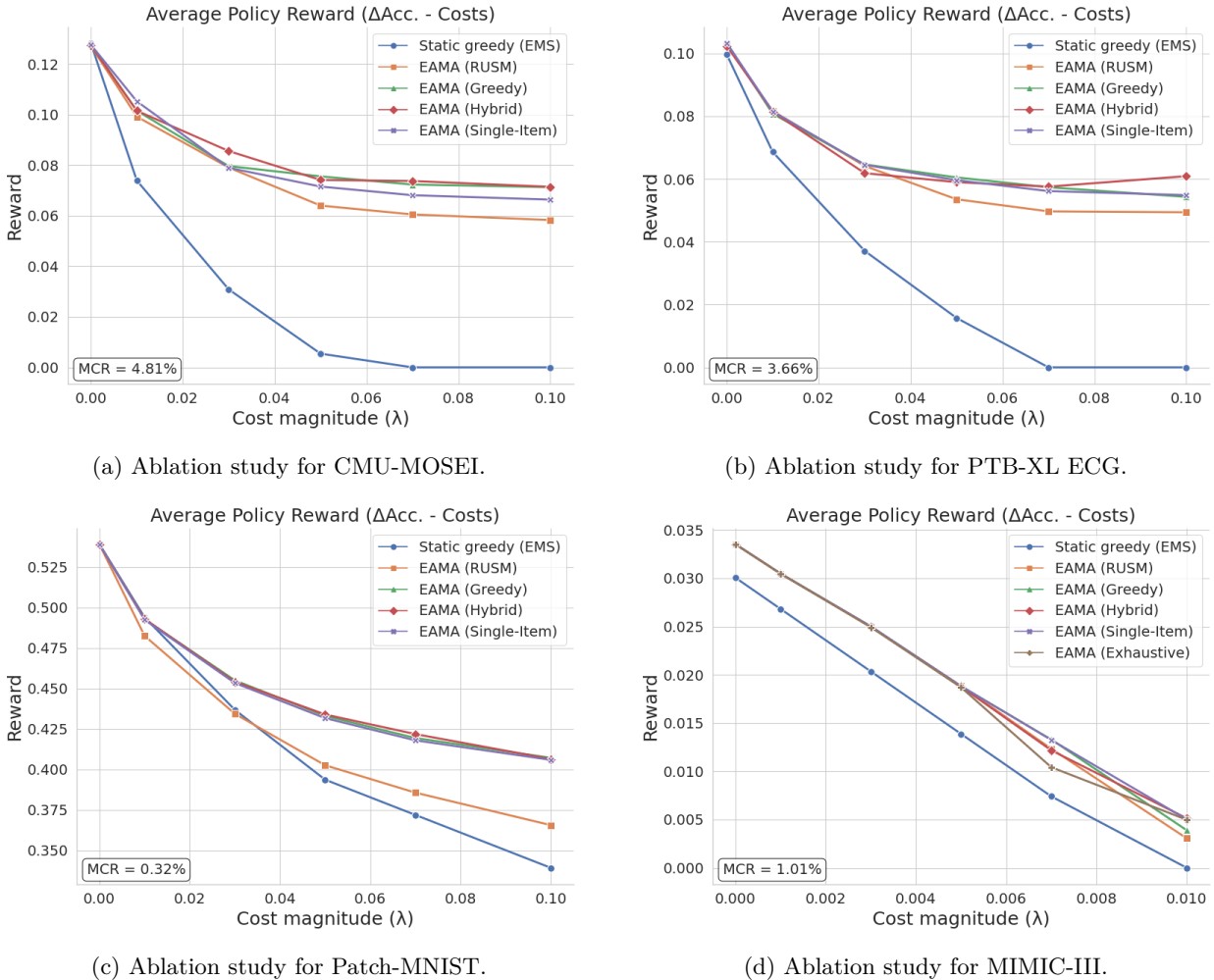

(a) Ablation study for CMU-MOSEI.

(b) Ablation study for PTB-XL ECG.

(c) Ablation study for Patch-MNIST.

(d) Ablation study for MIMIC-III.

Figure 11: We plot the average reward (Equation 10), defined as the accuracy gain from acquired modalities minus their acquisition costs, where costs are randomly sampled per input and scaled by the parameter $\lambda$. The bottom-left inset reports the empirical model confusion rate (MCR), measuring the probability that adding modalities changes a correct prediction to an incorrect one. This ablation study shows the varying performance of different optimization algorithms for maximizing the objective in Equation 3.

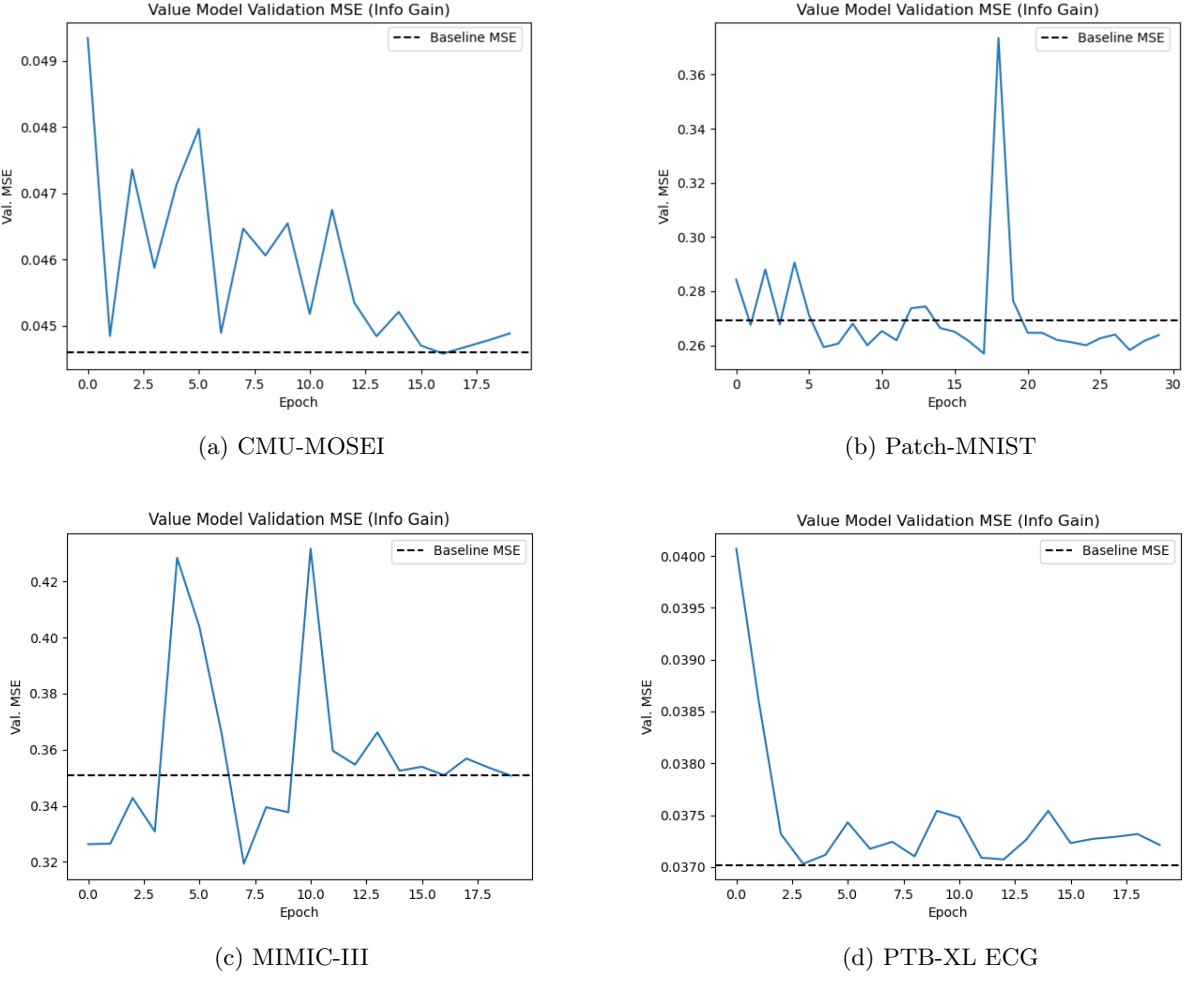

(a) CMU-MOSEI

(b) Patch-MNIST

(c) MIMIC-III

(d) PTB-XL ECG

Figure 12: Validation MSE for estimating the information gain value function. The baseline MSE corresponds to predicting the mean information gain value from the training dataset. In no experiments could the value model consistently outperform this baseline mean-value prediction.

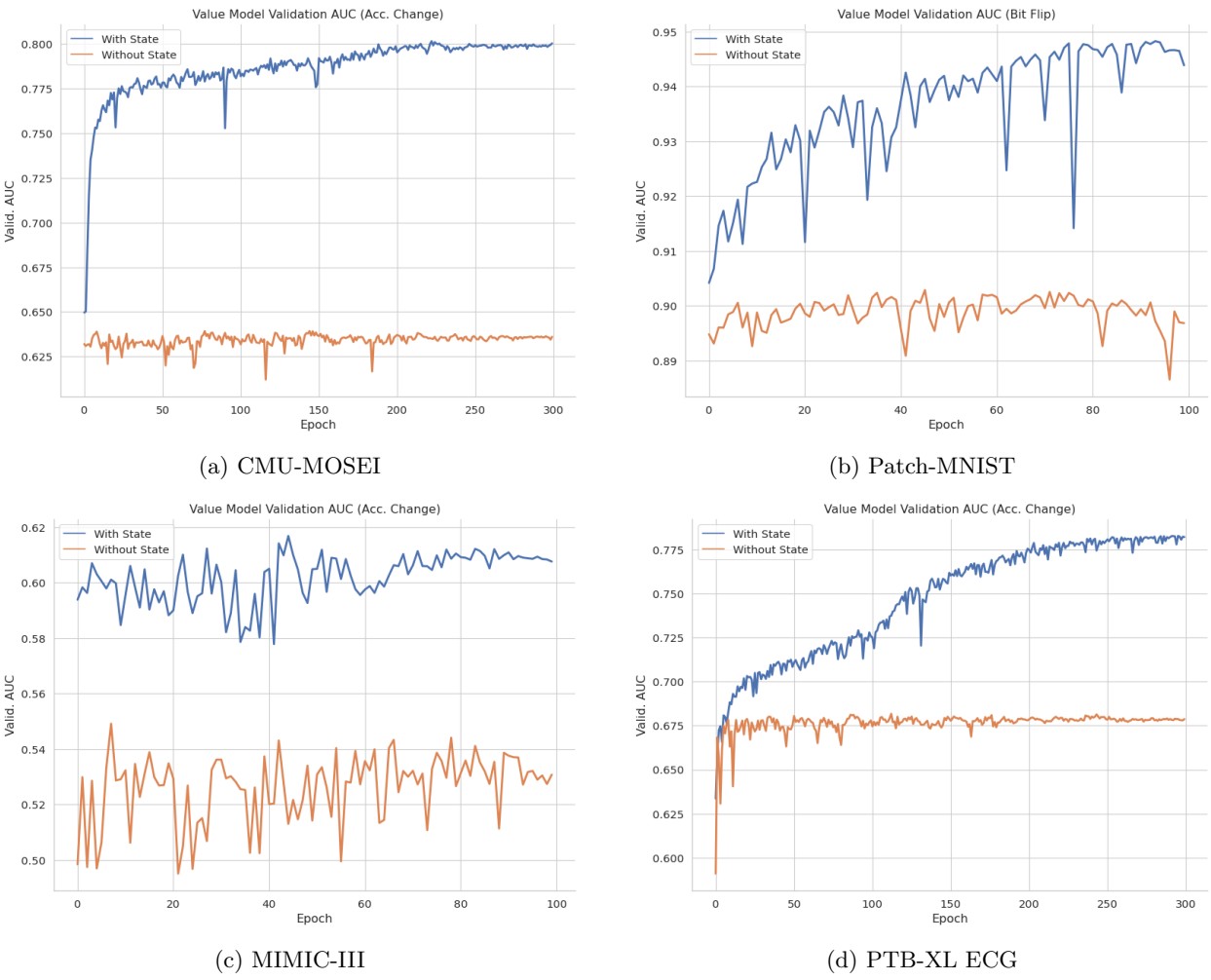

Figure 13: The ROC-AUC score on a validation dataset after each training epoch for value models trained with and without the state, $(\mathbf{c}, \mathbf{X}^p)$, as input. The value model trained without state can only observe the sets $(p, q)$ of acquired and selected modalities, but cannot see the observations themselves. These results show that the contents of the state are consistently important for predicting the value of unobserved modalities.

(a) $\lambda = 0$

| Method | Modalities Used (%) | Avg. Cost | Acc. (%) | Reward |
|---|---|---|---|---|
| Context-Only | 0.00 | 0.0000 | 69.06 | 0.0000 |
| EMS | 100.00 | 0.0000 | 79.03 | 0.0996 |
| RUSM | 50.65 | 0.0000 | 79.30 | 0.1024 |
| GreedySM | 47.87 | 0.0000 | 79.39 | 0.1033 |
| Hybrid | 50.16 | 0.0000 | 79.30 | 0.1024 |
| Single-Item | 47.94 | 0.0000 | 79.39 | 0.1033 |

(b) $\lambda = 0.01$

| Method | Modalities Used (%) | Avg. Cost | Acc. (%) | Reward |
|---|---|---|---|---|
| Context-Only | 0.00 | 0.0000 | 69.06 | 0.0000 |
| EMS | 18.18 | 0.0215 | 78.07 | 0.0686 |
| RUSM | 27.33 | 0.0203 | 79.25 | 0.0816 |
| GreedySM | 26.27 | 0.0175 | 78.89 | 0.0807 |
| Hybrid | 26.50 | 0.0178 | 78.98 | 0.0814 |
| Single-Item | 27.12 | 0.0186 | 79.07 | 0.0815 |

(c) $\lambda = 0.03$

| Method | Modalities Used (%) | Avg. Cost | Acc. (%) | Reward |
|---|---|---|---|---|
| Context-Only | 0.00 | 0.0000 | 69.06 | 0.0000 |
| EMS | 9.09 | 0.0320 | 75.98 | 0.0371 |
| RUSM | 21.37 | 0.0332 | 78.80 | 0.0641 |
| GreedySM | 20.42 | 0.0259 | 78.12 | 0.0646 |
| Hybrid | 20.45 | 0.0264 | 77.89 | 0.0619 |
| Single-Item | 20.26 | 0.0260 | 78.12 | 0.0645 |

(d) $\lambda = 0.05$

| Method | Modalities Used (%) | Avg. Cost | Acc. (%) | Reward |
|---|---|---|---|---|
| Context-Only | 0.00 | 0.0000 | 69.06 | 0.0000 |
| EMS | 9.09 | 0.0534 | 75.98 | 0.0157 |
| RUSM | 19.34 | 0.0384 | 78.25 | 0.0535 |
| GreedySM | 18.36 | 0.0282 | 77.93 | 0.0605 |
| Hybrid | 18.44 | 0.0279 | 77.75 | 0.0590 |
| Single-Item | 18.14 | 0.0282 | 77.84 | 0.0596 |

(e) $\lambda = 0.07$

| Method | Modalities Used (%) | Avg. Cost | Acc. (%) | Reward |
|---|---|---|---|---|
| Context-Only | 0.00 | 0.0000 | 69.06 | 0.0000 |
| EMS | 0.00 | 0.0000 | 69.06 | 0.0000 |
| RUSM | 17.91 | 0.0404 | 78.07 | 0.0497 |
| GreedySM | 16.49 | 0.0268 | 77.48 | 0.0574 |
| Hybrid | 16.54 | 0.0271 | 77.53 | 0.0575 |
| Single-Item | 16.34 | 0.0267 | 77.34 | 0.0561 |

(f) $\lambda = 0.10$

| Method | Modalities Used (%) | Avg. Cost | Acc. (%) | Reward |
|---|---|---|---|---|
| Context-Only | 0.00 | 0.0000 | 69.06 | 0.0000 |
| EMS | 0.00 | 0.0000 | 69.06 | 0.0000 |
| RUSM | 16.09 | 0.0393 | 77.93 | 0.0494 |
| GreedySM | 15.39 | 0.0253 | 77.02 | 0.0543 |
| Hybrid | 15.30 | 0.0260 | 77.75 | 0.0609 |
| Single-Item | 15.32 | 0.0257 | 77.12 | 0.0549 |

Table 3: Full results for the PTB-XL ECG experiments, using the accuracy change value function. The first row reports the accuracy when the classifier makes predictions with only the sample context. The second row reports the results when using Algorithm 4. The subsequent rows report the results for our EAMA method, as in Algorithm 1, using different optimization algorithms, outlined in Appendix B.

(a) $\lambda = 0$

| Method | Modalities Used (%) | Avg. Cost | Acc. (%) | Reward |
|---|---|---|---|---|
| Context-Only | 0.00 | 0.0000 | 45.01 | 0.0000 |
| EMS | 86.67 | 0.0000 | 98.93 | 0.5392 |
| RUSM | 100.00 | 0.0000 | 98.91 | 0.5390 |
| GreedySM | 100.00 | 0.0000 | 98.91 | 0.5390 |
| Hybrid | 100.00 | 0.0000 | 98.91 | 0.5390 |
| Single-Item | 100.00 | 0.0000 | 98.91 | 0.5390 |

(b) $\lambda = 0.01$

| Method | Modalities Used (%) | Avg. Cost | Acc. (%) | Reward |
|---|---|---|---|---|
| Context-Only | 0.00 | 0.0000 | 45.01 | 0.0000 |
| EMS | 20.00 | 0.0326 | 97.67 | 0.4940 |
| RUSM | 41.28 | 0.0363 | 96.90 | 0.4826 |
| GreedySM | 36.11 | 0.0272 | 97.01 | 0.4928 |
| Hybrid | 36.06 | 0.0270 | 97.02 | 0.4931 |
| Single-Item | 36.22 | 0.0274 | 97.00 | 0.4925 |

(c) $\lambda = 0.03$

| Method | Modalities Used (%) | Avg. Cost | Acc. (%) | Reward |
|---|---|---|---|---|
| Context-Only | 0.00 | 0.0000 | 45.01 | 0.0000 |
| EMS | 13.33 | 0.0650 | 95.20 | 0.4369 |
| RUSM | 34.10 | 0.0706 | 95.51 | 0.4344 |
| GreedySM | 30.59 | 0.0498 | 95.50 | 0.4551 |
| Hybrid | 30.52 | 0.0496 | 95.40 | 0.4543 |
| Single-Item | 30.76 | 0.0508 | 95.44 | 0.4535 |

(d) $\lambda = 0.05$

| Method | Modalities Used (%) | Avg. Cost | Acc. (%) | Reward |
|---|---|---|---|---|
| Context-Only | 0.00 | 0.0000 | 45.01 | 0.0000 |
| EMS | 6.67 | 0.0545 | 89.83 | 0.3937 |
| RUSM | 30.96 | 0.0901 | 94.29 | 0.4027 |
| GreedySM | 28.04 | 0.0612 | 94.45 | 0.4332 |
| Hybrid | 28.05 | 0.0616 | 94.57 | 0.4340 |
| Single-Item | 28.21 | 0.0629 | 94.48 | 0.4318 |

(e) $\lambda = 0.07$

| Method | Modalities Used (%) | Avg. Cost | Acc. (%) | Reward |
|---|---|---|---|---|
| Context-Only | 0.00 | 0.0000 | 45.01 | 0.0000 |
| EMS | 6.67 | 0.0762 | 89.83 | 0.3720 |
| RUSM | 28.98 | 0.1027 | 93.85 | 0.3857 |
| GreedySM | 26.36 | 0.0667 | 93.63 | 0.4195 |
| Hybrid | 26.35 | 0.0667 | 93.87 | 0.4219 |
| Single-Item | 26.52 | 0.0686 | 93.67 | 0.4180 |

(f) $\lambda = 0.10$

| Method | Modalities Used (%) | Avg. Cost | Acc. (%) | Reward |
|---|---|---|---|---|
| Context-Only | 0.00 | 0.0000 | 45.01 | 0.0000 |
| EMS | 6.67 | 0.1089 | 89.83 | 0.3393 |
| RUSM | 26.95 | 0.1125 | 92.83 | 0.3657 |
| GreedySM | 24.55 | 0.0690 | 92.63 | 0.4072 |
| Hybrid | 24.60 | 0.0694 | 92.60 | 0.4065 |
| Single-Item | 24.71 | 0.0710 | 92.71 | 0.4060 |

Table 4: Full results for the Patch-MNIST experiments, using the bit-flip value function. The first row reports the accuracy when the classifier makes predictions with only the sample context. The second row reports the results when using Algorithm 4. The subsequent rows report the results for our EAMA method, as in Algorithm 1, using different optimization algorithms, outlined in Appendix B.

(a) $\lambda = 0$

| Method | Modalities Used (%) | Avg. Cost | Acc. (%) | Reward |
|---|---|---|---|---|
| Context-Only | 0.00 | 0.0000 | 46.82 | 0.0000 |
| EMS | 100.00 | 0.0000 | 49.83 | 0.0301 |
| RUSM | 97.26 | 0.0000 | 50.17 | 0.0335 |
| GreedySM | 97.34 | 0.0000 | 50.17 | 0.0335 |
| Hybrid | 97.34 | 0.0000 | 50.17 | 0.0335 |
| Single-Item | 97.23 | 0.0000 | 50.17 | 0.0335 |

(b) $\lambda = 0.001$

| Method | Modalities Used (%) | Avg. Cost | Acc. (%) | Reward |
|---|---|---|---|---|
| Context-Only | 0.00 | 0.0000 | 46.82 | 0.0000 |
| EMS | 100.00 | 0.0032 | 49.83 | 0.0268 |
| RUSM | 95.41 | 0.0030 | 50.17 | 0.0305 |
| GreedySM | 95.45 | 0.0030 | 50.17 | 0.0305 |
| Hybrid | 95.53 | 0.0030 | 50.17 | 0.0305 |
| Single-Item | 95.22 | 0.0030 | 50.17 | 0.0305 |

(c) $\lambda = 0.003$

| Method | Modalities Used (%) | Avg. Cost | Acc. (%) | Reward |
|---|---|---|---|---|
| Context-Only | 0.00 | 0.0000 | 46.82 | 0.0000 |
| EMS | 100.00 | 0.0097 | 49.83 | 0.0203 |
| RUSM | 91.68 | 0.0086 | 50.17 | 0.0250 |
| GreedySM | 91.45 | 0.0085 | 50.17 | 0.0250 |
| Hybrid | 91.60 | 0.0085 | 50.17 | 0.0250 |
| Single-Item | 91.29 | 0.0085 | 50.17 | 0.0250 |

(d) $\lambda = 0.005$

| Method | Modalities Used (%) | Avg. Cost | Acc. (%) | Reward |
|---|---|---|---|---|
| Context-Only | 0.00 | 0.0000 | 46.82 | 0.0000 |
| EMS | 100.00 | 0.0162 | 49.83 | 0.0139 |
| RUSM | 88.63 | 0.0135 | 50.06 | 0.0188 |
| GreedySM | 89.02 | 0.0136 | 50.06 | 0.0188 |
| Hybrid | 89.09 | 0.0136 | 50.06 | 0.0188 |
| Single-Item | 88.79 | 0.0136 | 50.06 | 0.0188 |

(e) $\lambda = 0.007$

| Method | Modalities Used (%) | Avg. Cost | Acc. (%) | Reward |
|---|---|---|---|---|
| Context-Only | 0.00 | 0.0000 | 46.82 | 0.0000 |
| EMS | 100.00 | 0.0227 | 49.83 | 0.0074 |
| RUSM | 88.71 | 0.0189 | 49.94 | 0.0123 |
| GreedySM | 89.36 | 0.0191 | 50.06 | 0.0133 |
| Hybrid | 89.25 | 0.0190 | 49.94 | 0.0122 |
| Single-Item | 89.33 | 0.0191 | 50.06 | 0.0132 |

(f) $\lambda = 0.01$

| Method | Modalities Used (%) | Avg. Cost | Acc. (%) | Reward |
|---|---|---|---|---|
| Context-Only | 0.00 | 0.0000 | 46.82 | 0.0000 |
| EMS | 0.00 | 0.0000 | 46.82 | 0.0000 |
| RUSM | 83.35 | 0.0247 | 49.60 | 0.0031 |
| GreedySM | 84.39 | 0.0250 | 49.71 | 0.0039 |
| Hybrid | 84.35 | 0.0249 | 49.83 | 0.0051 |
| Single-Item | 84.39 | 0.0250 | 49.83 | 0.0050 |

Table 5: Full results for the MIMIC-III experiments, using the accuracy change value function. The first row reports the accuracy when the classifier makes predictions with only the sample context. The second row reports the results when using Algorithm 4. The subsequent rows report the results for our EAMA method, as in Algorithm 1, using different optimization algorithms, outlined in Appendix B.

(a) $\lambda = 0$

| Method | Modalities Used (%) | Avg. Cost | Acc. (%) | Reward |
|---|---|---|---|---|
| Context-Only | 0.00 | 0.0000 | 59.44 | 0.0000 |
| EMS | 75.00 | 0.0000 | 72.27 | 0.1283 |
| RUSM | 77.72 | 0.0000 | 72.16 | 0.1272 |
| GreedySM | 75.09 | 0.0000 | 72.20 | 0.1276 |
| Hybrid | 75.13 | 0.0000 | 72.18 | 0.1274 |
| Single-Item | 79.69 | 0.0000 | 72.20 | 0.1276 |

(b) $\lambda = 0.01$

| Method | Modalities Used (%) | Avg. Cost | Acc. (%) | Reward |
|---|---|---|---|---|
| Context-Only | 0.00 | 0.0000 | 59.44 | 0.0000 |
| EMS | 16.67 | 0.0215 | 68.98 | 0.0739 |
| RUSM | 41.93 | 0.0306 | 72.42 | 0.0992 |
| GreedySM | 41.60 | 0.0289 | 72.48 | 0.1015 |
| Hybrid | 41.13 | 0.0287 | 72.46 | 0.1015 |
| Single-Item | 41.71 | 0.0290 | 72.87 | 0.1052 |

(c) $\lambda = 0.03$

| Method | Modalities Used (%) | Avg. Cost | Acc. (%) | Reward |
|---|---|---|---|---|
| Context-Only | 0.00 | 0.0000 | 59.44 | 0.0000 |
| EMS | 16.67 | 0.0646 | 68.98 | 0.0308 |
| RUSM | 33.36 | 0.0496 | 72.31 | 0.0791 |
| GreedySM | 32.00 | 0.0424 | 71.64 | 0.0796 |
| Hybrid | 32.14 | 0.0431 | 72.31 | 0.0856 |
| Single-Item | 32.04 | 0.0436 | 71.69 | 0.0789 |

(d) $\lambda = 0.05$

| Method | Modalities Used (%) | Avg. Cost | Acc. (%) | Reward |
|---|---|---|---|---|
| Context-Only | 0.00 | 0.0000 | 59.44 | 0.0000 |
| EMS | 8.33 | 0.0538 | 65.36 | 0.0054 |
| RUSM | 29.63 | 0.0548 | 71.32 | 0.0640 |
| GreedySM | 28.61 | 0.0454 | 71.54 | 0.0755 |
| Hybrid | 28.57 | 0.0449 | 71.34 | 0.0741 |
| Single-Item | 28.70 | 0.0473 | 71.32 | 0.0715 |

(e) $\lambda = 0.07$

| Method | Modalities Used (%) | Avg. Cost | Acc. (%) | Reward |
|---|---|---|---|---|
| Context-Only | 0.00 | 0.0000 | 59.44 | 0.0000 |
| EMS | 0.00 | 0.0000 | 59.44 | 0.0000 |
| RUSM | 26.93 | 0.0556 | 71.04 | 0.0605 |
| GreedySM | 26.12 | 0.0437 | 71.04 | 0.0723 |
| Hybrid | 26.06 | 0.0428 | 71.09 | 0.0737 |
| Single-Item | 26.41 | 0.0467 | 70.91 | 0.0681 |

(f) $\lambda = 0.10$

| Method | Modalities Used (%) | Avg. Cost | Acc. (%) | Reward |
|---|---|---|---|---|
| Context-Only | 0.00 | 0.0000 | 59.44 | 0.0000 |
| EMS | 0.00 | 0.0000 | 59.44 | 0.0000 |
| RUSM | 24.77 | 0.0520 | 70.46 | 0.0583 |
| GreedySM | 23.76 | 0.0398 | 70.55 | 0.0713 |
| Hybrid | 23.74 | 0.0406 | 70.63 | 0.0714 |
| Single-Item | 23.92 | 0.0433 | 70.40 | 0.0664 |

Table 6: Full results for the CMU-MOSEI experiments, using the accuracy change value function. The first row reports the accuracy when the classifier makes predictions with only the sample context. The second row reports the results when using Algorithm 4. The subsequent rows report the results for our EAMA method, as in Algorithm 1, using different optimization algorithms, outlined in Appendix B.

