# OpenReview forum: "When is More Better? Efficient and Adaptive Modality Acquisition in Multimodal Learning"
_TMLR — Under review for TMLR_

### Review · Reviewer_JR2G · 2026-04-13

**Summary Of Contributions:**

This paper studies adaptive modality acquisition at the per-sample level, rather than choosing one fixed modality subset for the whole population. It learns a recursive energy-based value function over the currently observed modalities and a candidate subset to acquire, then approximately solves value-minus-cost subset selection via submodular optimization. A central conceptual contribution is the decomposition of accuracy-change value into a bit-flip term minus a model-confusion term, which motivates the proposed model confusion rate criterion for choosing between value functions. The method is evaluated on four datasets with experiments reporting favorable reward-cost trade-offs, sometimes improving accuracy while using only a fraction of the modalities.

**Audience:**

Yes

**Audience Explanation:**

This paper studies adaptive modality acquisition at the per-sample level, rather than choosing one fixed modality subset for the whole population. It learns a recursive energy-based value function over the currently observed modalities and a candidate subset to acquire, then approximately solves value-minus-cost subset selection via submodular optimization. A central conceptual contribution is the decomposition of accuracy-change value into a bit-flip term minus a model-confusion term, which motivates the proposed model confusion rate criterion for choosing between value functions. The method is evaluated on four datasets with experiments reporting favorable reward-cost trade-offs, sometimes improving accuracy while using only a fraction of the modalities.

**Claims And Evidence:**

Yes

**Claims Explanation:**

This paper studies adaptive modality acquisition at the per-sample level, rather than choosing one fixed modality subset for the whole population. It learns a recursive energy-based value function over the currently observed modalities and a candidate subset to acquire, then approximately solves value-minus-cost subset selection via submodular optimization. A central conceptual contribution is the decomposition of accuracy-change value into a bit-flip term minus a model-confusion term, which motivates the proposed model confusion rate criterion for choosing between value functions. The method is evaluated on four datasets with experiments reporting favorable reward-cost trade-offs, sometimes improving accuracy while using only a fraction of the modalities.

**Requested Changes:**

1.	The key optimization guarantee relies entirely on Assumption 3.1, namely that the expected value function is submodular for each state, yet the paper does not verify this assumption for the learned value model or for the actual datasets used in the experiments. Moreover, the final practical algorithm is a hybrid of RUSM and greedy, while only RUSM has the stated approximation guarantee. The paper moves from “assuming submodularity” to a fairly strong efficiency-and-quality narrative without quantifying how often the assumption is approximately true or how badly it fails.
2.	The manuscript argues that EMS is the only scalable baseline that can be reasonably evaluated, but this is not sufficient, especially because some of the experiments are in very small action spaces. In particular, for MIMIC-III the paper uses only three observable modalities, which is exactly the regime where stronger adaptive baselines, exhaustive search, dynamic programming, or carefully implemented RL-style policies should be feasible. Table 1 is not a substitute for actual experimental comparison, and without such comparisons it is difficult to assess whether the gains come from the proposed value formulation or simply from comparing against a weak static policy.
3.	Two of the four evaluation settings are heavily engineered in ways that weaken the claim of practical multimodal acquisition. Patch-MNIST treats image patches as “modalities,” and CMU-MOSEI is converted into many modalities by chunking precomputed text and facial embeddings into smaller pieces. These constructions are not convincing proxies for real modality-acquisition problems, where modalities are semantically distinct sources with distinct acquisition mechanisms and costs.
4.	The paper does not show robustness across different backbone classifiers, calibration qualities, or training schemes. I would expect a stronger analysis of whether the learned acquisition policy transfers across classifiers or whether it overfits to one model’s idiosyncrasies.
5.	The paper argues that one should prefer the bit-flip value when the expected confusion term is “approximately zero” and the accuracy-change value otherwise, yet no threshold selection rule, validation protocol, or statistical robustness analysis is provided. In practice, deciding whether an estimated MCR is small enough is itself a model-selection problem, and the paper currently treats that decision too heuristic.

# remarks
1.	The presentation of empirical results would be stronger with uncertainty estimates, significance testing, and clearer reporting of variability across random seeds.
2.	The manuscript should report training cost more carefully, not only inference-time cost. The paper highlights per-sample test-time optimization speed, but the training procedure for the value model involves repeated sampling over subset pairs, and the computational burden of that stage is not characterized with the same level of care.

---

> ### Author Response · Authors · 2026-06-17
>
> We thank the reviewer for the thoughtful and constructive feedback, and we are encouraged by the positive assessment of the paper's contribution to adaptive modality acquisition. In the revised manuscript, we made several changes that directly address the reviewer's concerns: we empirically validate the submodularity assumption for the learned value models, clarify the approximation guarantee of the hybrid RUSM/greedy optimizer, add a cost-adaptive EMS baseline and exhaustive-search comparison for MIMIC-III, clarify the role of constructed acquisition units in Patch-MNIST and CMU-MOSEI, expand the discussion of classifier-faithful value estimation and calibration, and add a clearer empirical guideline for choosing between the bit-flip and accuracy-change value functions. We address each point below.
>
> __1. Submodularity assumption and optimization guarantee.__
>
> We agree with the reviewer that the submodularity assumption is central to the optimization framework and should be directly examined for the learned value models used in the experiments. This was one of the main additions in the revised manuscript. We added an empirical validation of the submodularity assumption in Appendix C and updated the discussion in Section 3.2 to summarize these results. Specifically, for each final learned value model, we evaluate the second-order submodularity condition with respect to the proposed acquisition set. For context modality $\mathbf{c}$, observed subset $p$, proposed subset $q$, and modalities $i,j$ not already included, we compute
> $$\Delta_{ij}(\mathbf{c},\mathbf{X},q,p) = \widehat{Q}_f(\mathbf{c}, \mathbf{X}^{p},q\cup\{i,j\};\theta) - \widehat{Q}_f(\mathbf{c}, \mathbf{X}^{p},q\cup\{i\};\theta) - \widehat{Q}_f(\mathbf{c}, \mathbf{X}^{p},q\cup\{j\};\theta) + \widehat{Q}_f(\mathbf{c}, \mathbf{X}^{p},q;\theta).$$
>
> Submodularity requires $\Delta_{ij}\leq 0$. We estimate this quantity empirically over the test set by randomly sampling subsets $p$ and $q$ as in the value-model training procedure, and report the average second-order value $\bar{\Delta}$, its $95$% confidence interval, the positive violation rate $\mathrm{PVR}\_{1\sigma}$, and the statistically significant positive violation rate $\mathrm{SPVR}\_{0.05}$ after Benjamini--Hochberg correction across modality pairs.
>
> The results are summarized below:
> | Task | $\bar{\Delta}$ | $\bar{\Delta}\,[\ell_{0.95},u_{0.95}]$ | $\mathrm{PVR}_{1\sigma}$ (%) | $\mathrm{SPVR}_{0.05}$ (%) |
> |---|---:|---:|---:|---:|
> | PTB-XL EKG | -0.0026 | [-0.0032, -0.0022] | 0.59 | 0.0 |
> | Patch-MNIST | -0.0037 | [-0.0041, -0.0032] | 0.35 | 0.0 |
> | MIMIC-III | -0.0313 | [-0.0335, -0.0290] | 4.5 | 0.0 |
> | CMU-MOSEI | -0.0036 | [-0.0044, -0.0029] | 2.5 | 0.0 |
>
> Across all experimental settings, the average value of this second-order condition is negative and the upper bound of the $95$% confidence interval remains below zero. We also found zero statistically significant positive violations after Benjamini--Hochberg correction across modality pairs. These results provide direct empirical evidence that the learned value models used in our experiments satisfy the diminishing-returns structure required by the optimization framework.
>
> We also clarify the optimization guarantee for the final practical algorithm. The reviewer correctly notes that the final algorithm is a hybrid of RUSM and greedy optimization. However, the RUSM guarantee still applies to this hybrid procedure. The hybrid algorithm runs both RUSM and greedy, evaluates the objective value of both returned subsets, and selects the subset with the larger objective value. Therefore, the selected solution is always at least as good as the RUSM solution. In the worst case, the hybrid algorithm can do no worse than RUSM alone, so the $1/2$-approximation guarantee of RUSM is preserved. We have revised the manuscript to make this point explicit.

---

> > ### Author Response · Authors · 2026-06-17
> > **Part 2**
> >
> > Continuing our response:
> >
> > __2. Baseline comparisons and isolating the contribution of EAMA.__
> >
> > We agree with the reviewer that comparing only against a static EMS baseline does not fully isolate the contribution of our learned adaptive acquisition strategy. To address this concern, we added a new adaptive baseline in Section 4. Specifically, we extend the Efficient Modality Selection (EMS) baseline to obtain a \emph{cost-adaptive EMS policy}. The original EMS procedure constructs a greedy sequence of modality subsets $s_1,\dots, s_M$ using population-level accuracy gains. In the static EMS baseline, for each value of $\lambda$, we select the single subset in this sequence that maximizes the average reward. In the new adaptive EMS baseline, the policy instead adapts to the individual acquisition costs of each sample. Given sample-specific costs $\mathbf{z}_i$, the adaptive EMS policy selects
> >
> > $$\pi_{\mathrm{adapt}}(\lambda \cdot \mathbf{z}_i) = \arg\max\_{k\in[M]} a(\pi_k) - \lambda \cdot \mathbf{z}_i^\top \mathbf{s}_k,$$
> >
> > where $\mathbf{s}_k$ is the binary representation of the $k$-th greedy EMS subset. Thus, unlike static EMS, this baseline can acquire more modalities for samples with unusually low costs and fewer modalities for samples with unusually high costs. This comparison is designed to separate the benefit of EAMA from adaptivity alone. Adaptive EMS is sample-level adaptive, scalable, and cost-aware, but it does not use the currently observed modalities or context to estimate the sample-specific benefit of acquiring additional modalities. In contrast, EAMA uses the learned value model to reason about both the observed information and the candidate modalities to acquire. Therefore, the comparison isolates the benefit of context-aware value estimation beyond sample-specific cost adaptation.
> >
> > The results in Section 4 show that adaptive EMS consistently improves over static EMS, as expected, but still substantially underperforms EAMA. In some settings, EAMA achieves up to $6\times$ the average reward of adaptive EMS. Thus, the gains are not explained solely by comparing an adaptive method against a weak static policy; rather, they reflect the benefit of using a learned, adaptive, and context-aware acquisition policy.
> >
> > We also considered the reviewer's suggestion of RL-style adaptive policies. As the reviewer notes, such policies would require careful implementation in this setting; applying RL to AMA with heterogeneous acquisition costs, variable observed modality states, and combinatorial action spaces is not an off-the-shelf baseline, and would require substantial additional methodological development. We agree that this is an interesting direction for future work, but we focused this revision on adding a scalable cost-adaptive EMS baseline and, for the small-action-space MIMIC-III setting, an exhaustive-search EAMA optimizer.
> >
> > We also understand the reviewer's point that MIMIC-III has a small action space. To address this directly, we added exhaustive search as an additional optimization strategy for EAMA in Figure 9 of Appendix B, maximizing the learned objective over all possible modality subsets. We found no significant difference in average reward between exhaustive search and our hybrid RUSM/greedy optimization approach on MIMIC-III, likely because the action space is small enough that both approaches identify the same solutions. This result suggests that, in the small-action-space setting, the optimization routine itself is not driving the observed gains; rather, the key contribution remains the learned value model used to estimate sample-specific acquisition benefit from the observed context.

---

> > > ### Author Response · Authors · 2026-06-17
> > > **Part 3**
> > >
> > > Continuing our response:
> > >
> > > __3. Constructed modality settings in Patch-MNIST and CMU-MOSEI.__
> > >
> > > We thank the reviewer for raising this important point regarding the experimental setup. We agree that Patch-MNIST and CMU-MOSEI involve constructed acquisition units. Our goal with these settings is not to claim that every constructed unit corresponds to a distinct real-world sensing modality, but rather to evaluate whether the proposed EAMA pipeline can efficiently solve adaptive acquisition problems with many correlated sources of information.
> > >
> > > For Patch-MNIST, we note that this setup has been used in prior work, notably by He et al. (2024), where the same dataset and acquisition setup are used as one of the main experiments for evaluating static efficient modality selection. While image patches are not distinct semantic modalities in the strict sense, they are disjoint, correlated feature groups corresponding to the same underlying sample and provide a controlled benchmark for evaluating acquisition policies. This experiment also plays a useful diagnostic role in our paper: Patch-MNIST is a relatively low-noise task for which the model confusion rate is very small, and the results show that this is precisely the regime where the bit-flip value function outperforms the accuracy-change value function. Thus, the experiment helps validate the connection between MCR and the choice of value function.
> > >
> > > For CMU-MOSEI, the underlying data are genuinely multimodal, including language, visual, and acoustic information. To create a larger-scale acquisition problem, we partition precomputed representations within these modalities into multiple chunks. We agree that these chunks should be interpreted as acquirable views or information units rather than independent physical modalities. However, they still represent distinct subspaces of the same multimodal sample, containing correlated but non-identical information that can be acquired and combined to improve prediction. This construction allows us to evaluate the scalability of EAMA in a setting with many candidate information sources while retaining a connection to a standard multimodal benchmark.
> > >
> > > More broadly, our experimental suite is intended to cover both controlled acquisition benchmarks and more application-facing multimodal settings. Patch-MNIST and the chunked CMU-MOSEI setup stress-test scalability and the behavior of different value functions, while the remaining experiments evaluate the method in settings with more naturally distinct acquisition units.
> > >
> > > __4. Robustness to classifier choice, calibration, and training scheme.__
> > >
> > > We agree that robustness across backbone classifiers and training schemes is an interesting and important direction for future work. In the present paper, however, our setting assumes a fixed downstream predictor, and the goal of the value model is to estimate the expected benefit of acquiring additional modalities for that specific predictor. In this sense, the value model is faithful to the classifier.
> > >
> > > More specifically, the ``value'' of acquiring a modality in our setting depends on both the information contained in the underlying data and the deployed classifier's ability to use that information to improve its prediction. A predictor-independent value model might appear more transferable across classifiers, but it could also recommend modalities that are informative in principle but not useful for the specific classifier being deployed. By contrast, a classifier-faithful value model recommends modalities that are both informative and usable by the downstream model. This is also central to our discussion of model confusion: a partial view may be confusing because of the data distribution itself, because of the classifier's behavior on that partial view, or both. For deployment, we therefore believe it is appropriate for the value model to be faithful to the specific classifier whose acquisition decisions it supports. That said, we agree that the underlying predictor should not be trained in a way that makes acquisition decisions brittle or dominated by spurious modality-specific behavior. To mitigate this, we use modality dropout when training the classifier, which encourages the predictor to remain robust to different observed modality subsets and reduces overreliance on any particular modality or feature group.
> > >
> > > We also added an analysis of post-hoc value-model calibration in Appendix B. Specifically, we evaluate temperature scaling for the learned value model and report calibration plots before and after calibration, together with the corresponding reward curves, for CMU-MOSEI. Across our experiments, we found that post-hoc calibration had negligible effect on the final acquisition reward, suggesting that the acquisition policy is not highly sensitive to this calibration adjustment in our setting. We therefore include this analysis in the appendix rather than as a main experimental result.

---

> > > > ### Author Response · Authors · 2026-06-17
> > > > **Part 4**
> > > >
> > > > Continuing our response:
> > > >
> > > > __5. MCR criterion and value-function selection.__
> > > >
> > > > We thank the reviewer for raising this point. We agree that the original manuscript should more clearly explain how the model confusion rate (MCR) is intended to guide the choice between the bit-flip and accuracy-change value functions. In the revised manuscript, we added a discussion in Section 4 under the subsection "MCR Influences Design and Performance" to make this guideline explicit and to connect it to the empirical behavior observed across experiments.
> > > >
> > > > Our results suggest that when MCR is very small, approximately below $1$% in our experiments, the bit-flip value function is preferable. In this regime, the model confusion term is negligible, so the bit-flip value closely approximates the accuracy-change value while being easier to learn. Another way to view this setting is as an imbalanced classification problem for the value model: when confusion events are extremely rare, the accuracy-change target contains very few examples of negative flips, which can degrade the learned value model. This helps explain why the bit-flip value function performs better in low-MCR settings such as Patch-MNIST.
> > > >
> > > > When MCR is non-negligible, however, the bit-flip value can overestimate the benefit of acquisition because it does not account for cases where adding modalities flips a correct prediction to an incorrect one. In this regime, the accuracy-change value function is more appropriate because it explicitly subtracts the model-confusion term and therefore better matches the actual acquisition objective.
> > > >
> > > > We have revised the main text to present this as an empirical model-selection guideline rather than as a universal theoretical threshold. In practice, if the MCR is clearly very small, our results support using the bit-flip value function; otherwise, the accuracy-change value function is safer because it directly estimates expected accuracy change. Practitioners can also train both value models and select between them using validation reward. As reported in Section 4, the longest value-model training time across all experiments is less than 10 hours, so training both value models sequentially would still take less than a day in our experimental setup, and could be faster if run in parallel. Since this is a one-time offline training cost, this provides a practical fallback when the MCR criterion is ambiguous. We agree that developing a more formal threshold-selection rule with statistical uncertainty estimates would be valuable future work, but the revised manuscript now makes the empirical criterion and its connection to value-model learnability clearer.
> > > >
> > > > __Additional remarks: uncertainty estimates and training cost.__
> > > >
> > > > We also thank the reviewer for the suggestions regarding uncertainty estimates and clearer reporting of training cost. We agree that a full multi-seed uncertainty analysis would further strengthen the empirical evaluation. Due to the computational cost of repeating the complete value-learning and acquisition pipeline across all datasets, we did not add a full multi-seed analysis in this revision. However, we added uncertainty and statistical testing for the central methodological assumption in the paper: the submodularity of the learned value models, including $95$% confidence intervals and multiple-testing-corrected violation tests. We also added value-model training curves in Appendix B and now report hardware specifications, seconds per epoch, total number of epochs, and wall-clock training time for each experiment in Section 4. These additions better characterize both the stability and computational cost of value-model training.
> > > >
> > > > We thank the reviewer again for their thoughtful comments and feedback, which have improved the quality of the revised manuscript.

---

### Review · Reviewer_bMPC · 2026-04-26

**Summary Of Contributions:**

The authors develop "efficient adaptive modality acquisition" for the task of choosing a sequence of input modalities (with varying costs) per sample. They set up a reasonable framing, including assuming submodularity of the value function. They present promising results on various datasets, although the costs are synthetic.

**Audience:**

Yes

**Audience Explanation:**

This is a relevant framing for real-world application of ML models.

**Broader Impact Concerns:**

None.

**Claims And Evidence:**

Yes

**Claims Explanation:**

Both the setup and experiments are very clearly presented. The experiments are thorough covering a range of simpler tabular data, imaging and natural language inputs. Ablations are included.

**Requested Changes:**

All relatively minor:
1) I think it would simplify the exposition to remove the context "c" and just treat it as an initially selected modality.
2) $\hat{Q}_{f} : S \times {0, 1}^M \rightarrow R$. This is a slight abuse of notation, it should be $\hat{Q}_{f} : S \times [M] \rightarrow R$
3) I disagree that $||e_p||_2$ encodes the size of the subset (maybe if $e_p$ were orthonormal it would)
4) Comparison to attention based MIL for aggegrating the e in q and p would be valuable: straight averaging doesn't feel very expressive.
5) I agree with the logic of uniformly sampling k1 and k2 but would be curious if even greater bias towards smaller values would be beneficial (since that is presumably usually the regime of interest)

Top of page 5, p ⊆ M should be p ⊆ [M].
Lemma 1 has an extra ) in the last line.

---

> ### Author Response · Authors · 2026-06-17
>
> We thank the reviewer for the positive feedback and for the helpful suggestions. We are encouraged that the reviewer found the setup and experiments clear and viewed the paper as relevant to real-world applications of machine learning. In the revised manuscript, we addressed the requested changes by clarifying the context-modality notation, correcting and explaining the value-model signature, adding an empirical analysis of the learned modality embeddings, discussing attention-based MIL aggregation as future work, clarifying the subset-size sampling distribution, and fixing the noted typographical issues. We respond to each point below.
>
> __1. Context modality notation.__
> We thank the reviewer for this suggestion. We agree that the original use of ``context'' could make $\mathbf{c}$ appear to be a separate object from the modality set, which was not our intention. In the revised manuscript, we now refer to $\mathbf{c}$ consistently as the \emph{context modality} to clarify that it is simply an initially observed modality or source of information. We keep this notation because the AMA setting assumes that some information is available before acquisition begins, although this context modality may be minimal or uninformative in some applications. Thus, $\mathbf{c}$ should be interpreted as part of the observed information available at the initial state, rather than as a separate modeling primitive.
>
> __2. Function signature of the value model.__
> We thank the reviewer for raising this point. We clarify that the value model is intended to score a proposed \emph{subset} of modalities, not a single candidate modality. Thus, the input is the current state $s\in\mathcal{S}$ together with a binary vector in $\{0,1\}^M$ representing the proposed subset of modalities to acquire. The function signature $\widehat{Q}_f : \mathcal{S} \times \{0,1\}^M \rightarrow \mathbb{R}$ is therefore the intended one. A signature of the form $\mathcal{S}\times [M]\rightarrow \mathbb{R}$ would correspond to scoring only one candidate modality at a time, whereas our method allows the value model to evaluate arbitrary proposed acquisition subsets. We have clarified the surrounding notation to make this distinction more explicit.
>
> __3. Norm of the averaged modality embedding.__
> We thank the reviewer for pointing this out. We agree that the norm of the averaged modality embedding does not generally encode subset size without additional assumptions on the geometry of the learned embeddings. For example, if all unit-norm modality embeddings were identical, then the norm of the averaged subset embedding would remain constant for every subset size. We have revised the manuscript to avoid suggesting that this relationship holds automatically.  Because this question is important for our mean-pooling set encoder, we added a new analysis in Appendix C, titled ``Learning to Encode Modality Subset Size.'' In this analysis, for each experiment, we sample modality subsets of different cardinalities and measure the average norm of the corresponding mean-pooled modality embedding. We compare these curves to the idealized orthogonal case, where for unit-norm embeddings,
>
> $$ ||\mathbf{e}_p||_2^2 = \frac{1}{|p|} + \frac{1}{|p|^2}  \sum\_{i \neq j} \mathbf{e}_i^\top \mathbf{e}_j,$$
>
> so if the embeddings are orthogonal, $||\mathbf{e}_p||_2 = 1/\sqrt{|p|}$.
>
> Empirically, we find that the learned embeddings roughly follow this expected decreasing curve, and all subset-norm curves are monotone decreasing in subset size. We also measure the average cosine similarity between learned modality embeddings and find no strong positive alignment: $-0.09$ for MIMIC-III, $0.05$ for PTB-XL EKG, $0.0015$ for CMU-MOSEI, and $0.03$ for Patch-MNIST. Thus, while mean pooling does not mathematically guarantee recovery of subset size in general, the learned modality embeddings in our experiments are sufficiently diverse that subset cardinality can be approximately recovered from the pooled representation. We also note in Appendix C that more expressive set encoders, such as attention-based multiple instance learning, are a promising direction for future work.

---

> > ### Author Response · Authors · 2026-06-17
> > **Part 2**
> >
> > Continuing our response:
> >
> > __4. Attention-based MIL aggregation.__
> > We agree that attention-based multiple instance learning (MIL) is a promising direction for constructing more expressive set encoders for observed and proposed modality subsets. In the revised manuscript, we now mention this direction in Appendix C in the discussion of learned modality embeddings. Our current mean-pooling encoder was chosen because it is simple, permutation-invariant, computationally efficient, and empirically sufficient for demonstrating the effectiveness of EAMA across the experiments considered. That said, we agree that an MIL-style encoder could be valuable in future work. In the AMA setting, such an encoder would likely need to go beyond directly applying standard MIL aggregation, since the representation of an observed modality may ideally depend not only on the other observed modalities in the set, but also on the proposed modalities being evaluated for acquisition.
> >
> > __5. Sampling distribution over observed and proposed subset sizes.__
> > We thank the reviewer for this suggestion. We agree that biasing the sampling distribution toward smaller values of $k_1$ and $k_2$ could be beneficial in settings where the acquisition policy is expected to operate primarily in small-subset regimes. In the current paper, we sample $k_1$ and $k_2$ uniformly to provide broad coverage over the possible observed and proposed subset sizes encountered by the recursive value model. This design choice is intended to avoid over-specializing the value model to only one part of the acquisition process. Since EAMA may be queried at different stages of acquisition, the value model should remain accurate not only when few modalities have been acquired, but also after several modalities have already been observed. This can be especially important in low-cost regimes, where the optimal policy may acquire most modalities and only drop a few, or acquire all modalities for many samples. Thus, covering both small and large subset sizes is useful in general. We agree that the sampling distribution could be adapted when the deployment regime is known in advance. For example, if an application is known to operate almost exclusively under high acquisition costs, then biasing toward smaller observed and proposed subsets may improve value-model accuracy in the most relevant region of the state space. We have added discussion of this tradeoff in Section 3.3 and view task-specific subset-size sampling distributions as a useful direction for future work.
> >
> > __6. Notation and typographical corrections.__
> > We thank the reviewer for catching these issues. We have corrected the notation on page 5 so that modality subsets are written as $p\subseteq [M]$, with $[M]$ denoting the set of modality indices and $M$ denoting the number of modalities. We have also fixed the extra parenthesis in the final line of Lemma 1.
> >
> > We thank the reviewer again for their feedback and comments, which have helped improve the revised manuscript.

---

### Review · Reviewer_C1nG · 2026-06-05

**Summary Of Contributions:**

Summary
This paper studies the problem of adaptive modality acquisition, where the goal is to determine which modalities should be acquired for an individual sample in order to balance predictor performance and modality acquisition cost. More specifically, given a predictor and a partially observed sample, the method aims to identify additional modalities that are worth acquiring before making the final prediction. To be fair, I believe the proposed method provides a workable solution to this problem, and the experimental results suggest that it performs well empirically across several benchmark datasets. The problem itself is meaningful and practically relevant, particularly in applications where acquiring additional modalities incurs non-negligible costs. At the same time, as discussed in the weaknesses section, I have concerns regarding the novelty of the proposed framework, the motivation behind several key design choices, and the rigor of some of the paper's claims and interpretations.

Strength
1.	I believe the paper studies an important and practically relevant problem. In the era of large foundation models, our ability to further improve performance by modifying the underlying model architecture is often limited by computational cost, training resources, and deployment constraints. In contrast, adaptively acquiring additional information for individual samples provides a potentially economical way to improve predictive performance without retraining or redesigning the predictor itself. For this reason, I view adaptive modality acquisition as a promising research direction with substantial practical potential.

2.	I also believe the paper makes a useful effort toward formalizing the adaptive modality acquisition problem. Although I have several concerns regarding specific design choices and aspects of the mathematical formulation, the overall framework clearly defines the problem setting, the acquisition objective, and the interaction between modality value estimation and modality selection. In this sense, the paper provides a structured foundation that future work can refine, extend, or build upon.

Weakness:
1.Validity of Q.(does Q measures the quantity that the author claimed to measure?)
A fundamental concern lies in the validity of the proposed value function itself. The paper motivates (Q) as a measure of the value of acquiring additional modalities. However, the supervision signals introduced in Sec. 3.4 (e.g., (Q^A), (Q^B), and (Q^C)) are highly discretized quantities derived from prediction outcome transitions. In particular, these targets only indicate whether the acquisition of a modality changes the prediction outcome, rather than quantifying the magnitude of the resulting benefit.
As a consequence, the supervision process does not explicitly teach the value model to distinguish between modalities with different levels of utility. Two modalities that provide substantially different predictive improvements may receive exactly the same target value as long as they induce the same prediction transition. Therefore, it is unclear whether the learned function is estimating modality value in the sense described throughout the paper, or merely predicting the likelihood that acquiring a modality will alter the classifier's decision.
This issue is further obscured by the presentation of continuous-valued outputs such as (\hat{Q}=0.07). Such values may give the impression that the method has learned a continuous utility measure. However, since the underlying supervision signals are discrete and only encode outcome transitions, a quantity such as 0.07 appears more naturally interpretable as an expected transition score (or the probability of obtaining a beneficial prediction change) rather than a direct estimate of modality utility. The distinction is important because these interpretations correspond to substantially different notions of “value”.
The authors may argue that their notion of value is intentionally defined through prediction outcome changes rather than through a conventional utility measure. Such a design choice is reasonable from an engineering perspective. However, this interpretation should be stated much more explicitly, as the current presentation may lead readers to believe that the method estimates a richer notion of modality utility than what is actually encoded in the training targets.
More importantly, the proposed targets appear too coarse to capture different degrees of acquisition benefit. As defined, they generally cannot distinguish between candidate modalities that produce different magnitudes of improvement, making it unclear whether they are sufficient to support the paper’s broader claims regarding modality value estimation.

2. theory-method consistency concern.
This concern relates to the connection between the proposed value functions and the submodular optimization framework.
The paper argues that efficient modality acquisition can be achieved by optimizing a submodular value function, which is a well-established and reasonable strategy to me. Indeed, much of the theoretical development in Section 3.2 and Appendix A is devoted to showing that, under Assumption 3.1, the acquisition objective becomes a non-monotonic submodular function and can therefore be approximately optimized using efficient greedy-style algorithms.
However, the critical issue is that the paper does not appear to establish that the proposed value functions themselves ((Q_f^A), (Q_f^B), or (Q_f^C)) satisfy the required submodularity assumption. Appendix A proves several properties of these value functions and their estimators, but I could not find either a theoretical justification or an empirical verification that the underlying value functions are submodular. Instead, submodularity is simply assumed in Assumption 3.1 and subsequently used to derive the optimization guarantees.
This creates a potential logical gap in the framework. The two central components of the method are (i) value estimation and (ii) modality subset optimization. The optimization procedure relies critically on the estimated value function being submodular, yet the paper does not establish that the proposed value formulations possess this property. As a result, it remains unclear whether the theoretical guarantees obtained from the submodular optimization analysis are actually applicable to the value functions used in practice.
This issue is particularly important because the proposed value functions are defined through classifier prediction changes. In many multimodal settings, complementary modalities may exhibit strong interaction effects, where the utility of acquiring a modality can increase rather than decrease as more modalities become available. Such behavior is not obviously consistent with the diminishing-returns property required by submodularity. Therefore, additional discussion, justification, or empirical evidence is needed to clarify whether the assumptions underlying the optimization framework are satisfied by the proposed value estimators.

3.The justification of Q being submodular.
The next concern is the justification for Assumption 3.1 itself. The paper motivates the submodularity assumption by citing prior empirical observations that modality acquisition exhibits diminishing marginal returns. However, it is unclear what notion of "return" these observations actually refer to, and more importantly, how they relate to the value functions proposed in this work.
The subsequent framework does not directly optimize or estimate a generic notion of information gain or utility. Instead, the value model is trained using specifically designed targets such as (Q_f^A), (Q_f^B), and (Q_f^C), which are defined through prediction outcome transitions. The paper does not establish a connection between the empirical diminishing-return observations used to motivate Assumption 3.1 and these particular value formulations.
As a result, the logical basis of the assumption remains unclear. Even if some notion of modality return is empirically observed to exhibit diminishing returns, it does not immediately follow that the proposed value functions, or their learned estimators, inherit the same property. The current argument appears to rely on transferring a structural property observed for one quantity to a different quantity without formally establishing the relationship between them.
I believe the paper would benefit from a more explicit discussion of how the motivating notion of modality return relates to the proposed value functions. Without such a connection, it is difficult to assess whether Assumption 3.1 is merely a convenient optimization assumption or a property that is genuinely supported by the value formulations introduced in the paper.

4.The cost of training Q estimator and its convergence condition are not discussed in the experiment.
The paper emphasizes reducing modality acquisition costs, yet the proposed solution introduces an additional value-learning stage whose supervision requires repeatedly sampling modality subsets and evaluating the underlying classifier. The manuscript provides little analysis of the amount of supervision required to train the value estimator, how performance scales with the number of sampled subset pairs, or whether the resulting training overhead is justified by the acquisition savings obtained at deployment. Since value estimation is a core component of the framework, understanding its sample complexity and training cost seems important for evaluating the practical efficiency of the proposed approach.

5.The treatment of modality acquisition costs and their interaction with the proposed value function is not clear. The practical use of the proposed method is limited for any real-world cost scenarios.
The optimization objective directly subtracts acquisition cost from the estimated modality value. However, it is not obvious that these two quantities are naturally comparable. The estimated value function is intended to measure the benefit of acquiring additional modalities, while acquisition costs may represent fundamentally different concepts depending on the application. For example, in healthcare, one modality may incur monetary cost (e.g., the price of an MRI scan), another may incur waiting time, while a third may involve medical risk or patient discomfort. These quantities do not necessarily share a common unit or interpretation.
The paper implicitly assumes that all such factors can be mapped into a single scalar cost and directly traded against the estimated modality value through a fixed coefficient (\lambda). However, the manuscript provides little discussion of how such a mapping should be performed in practice. For example, what does it mean to trade a unit increase in estimated modality value against one unit of acquisition cost? How should a practitioner calibrate (\lambda)? More importantly, if different modalities incur qualitatively different forms of cost (e.g., dollars versus medical risk), it is unclear how these should be represented within the proposed framework.
This issue becomes particularly important because the acquisition policy is entirely determined by this trade-off. If a modality is assigned an excessively large cost, the optimization procedure may effectively never select it, regardless of its predictive utility. Conversely, if a modality is assigned a very small cost, it may be selected for nearly every sample. In both cases, the resulting behavior is driven more by cost calibration than by the learned value model itself.
For this reason, I would have liked to see a more detailed analysis of the resulting acquisition behavior. For example, reporting modality selection frequencies, modality usage distributions, or how acquisition patterns change under different cost settings would provide valuable insight into whether the method is genuinely learning sample-specific modality utility or simply reflecting manually assigned cost scales.
More broadly, the experiments appear to use synthetically generated scalar costs rather than application-specific acquisition costs. While this is understandable for benchmarking purposes, it makes it difficult to assess how the proposed value-cost trade-off would operate in realistic deployment scenarios where costs may be heterogeneous, multidimensional, and not directly comparable.

6.Comparison in the experiment does not fully support the central claim of the paper.
I also have some questions regarding the experimental baselines. The paper states that EMS was selected because it was the only scalable approach that could be reasonably evaluated under heterogeneous acquisition costs. While this is a reasonable practical consideration, I would appreciate additional discussion regarding the methods that were excluded and the specific reasons why they were considered unsuitable for comparison.
More importantly, I am not fully convinced that the current experimental setup isolates the contribution of the proposed adaptive modality acquisition strategy itself. As I understand it, EMS is fundamentally a static modality acquisition method that selects a population-level subset of modalities, whereas EAMA performs sample-specific acquisition. Consequently, the comparison primarily demonstrates the benefit of adaptive acquisition relative to static acquisition.
However, this does not necessarily establish the effectiveness of the proposed acquisition policy. Even in the absence of existing adaptive baselines that satisfy all experimental constraints, I believe it would be informative to compare EAMA against simple sample-level alternatives. For example, one could randomly acquire the same number of modalities selected by EAMA for each sample and compare the resulting performance. Such an experiment would help determine whether the gains arise from the learned value estimator and acquisition strategy itself, rather than from adaptivity alone.
Overall, I believe the current EMS comparison is valuable, but additional baselines would provide a clearer understanding of what component of the framework is responsible for the observed improvements.

7.Some presentation issues in problem formalization
I also found the formalization and notation throughout the paper unnecessarily difficult to follow, with several definitions appearing inconsistent with the notation introduced earlier in the manuscript.
For example, Eq. (1) defines (p \subseteq M), although (M) was previously introduced as the number of modalities rather than a set. Based on the surrounding notation, it seems that (p \subseteq [M]) was intended. Similarly, the use of (c) in Eq. (1) appears inconsistent with the earlier definitions. According to the notation section, (c) is an indicator/index rather than a concrete set-valued object, making the union operation difficult to interpret. If the intention was to refer to the views associated with class (c), then the notation would appear to require something closer to (n_c) or ([n_c]), rather than (c) itself. As written, the expression is difficult to reconcile with the preceding definitions.
I was also confused by the treatment of views throughout the paper. The formulation initially introduces views as instance-specific quantities where the view is clearly an attribute of a particular sample. However, after the initial setup, the instance index (i) disappears entirely and the paper switches to expressions such as (X^p) and (X^q). Since the proposed problem is sample-level adaptive modality acquisition rather than population-level modality selection, it is not obvious what these quantities represent without explicitly binding them to a particular instance. While this may be intended as a shorthand notation, the transition is never explained and makes the formal development harder to follow.
A more substantial issue arises in the expectation operator in Eq. (2). This expectation is taken over both the label (y) and the unobserved view (X^q), conditioned on the currently observed information ((c,X^p)). However, the paper never explicitly discusses the conditional distribution (P(y,X^q| c,X^p)) underlying this expectation. While such a distribution may in principle be induced by the data-generating process over complete multimodal samples, the connection is left entirely implicit. This point is particularly important because the original problem formulation is defined over the joint distribution of complete multimodal samples ((X,y)), whereas Eq. (2) suddenly requires a conditional distribution involving only a subset of views. At a minimum, I would have expected some discussion of how the complete-data distribution induces the conditional distribution used in the value function, or what assumptions are being made regarding the relationship between observed and unobserved views. Without such clarification, the probabilistic interpretation of the expected value function remains somewhat unclear.
None of these issues fundamentally prevent the reader from understanding the overall methodology. However, they introduce unnecessary ambiguity and make it considerably more difficult to carefully verify the mathematical formulation and the assumptions underlying the proposed framework.

**Audience:**

Yes

**Audience Explanation:**

Yes. I think adaptive modality acquisition is an important and increasingly relevant problem, particularly in settings where acquiring additional information incurs non-trivial cost. The paper proposes a concrete framework for studying this problem and reports encouraging empirical results. Even though I have concerns regarding several aspects of the methodology and evaluation, I believe researchers interested in multimodal learning, cost-sensitive prediction, and adaptive information acquisition would find the problem formulation and findings of interest.

**Claims And Evidence:**

No

**Claims Explanation:**

Not entirely. While the empirical results are promising, I do not believe the current evidence fully supports all of the paper's central claims. In particular, the experimental comparison mainly demonstrates the benefit of adaptive acquisition over static acquisition, but does not fully isolate the contribution of the proposed value estimation framework itself. I would also have liked to see stronger evidence regarding the training requirements of the value estimator and the practical implications of the proposed value-cost tradeoff.

**Requested Changes:**

-Clarify the interpretation of the proposed value function. In particular, the paper should more explicitly discuss the relationship between the proposed targets ((Q_f^A), (Q_f^B), (Q_f^C)) and the notion of modality value advocated throughout the paper, including the implications of using highly discretized supervision signals.
-Better justify the submodularity assumption. The paper should either provide theoretical or empirical evidence that the proposed value functions (or their estimators) satisfy the properties required by the optimization framework, or clearly discuss the limitations of this assumption.
-Strengthen the experimental validation of the acquisition strategy. In addition to EMS, I would like to see comparisons that better isolate the contribution of the proposed value estimation framework, such as simple sample-level acquisition baselines with matched acquisition budgets.
-Provide additional discussion regarding the practical cost-value tradeoff and the training requirements of the value estimator, including the amount of supervision required, training overhead, and the interpretation of acquisition costs in realistic applications.
-Improve the clarity and consistency of the mathematical formulation and notation, particularly in the problem setup and value function definitions.

---

> ### Author Response · Authors · 2026-06-17
>
> We thank the reviewer for the detailed and constructive feedback. We are encouraged that the reviewer views adaptive modality acquisition as an important and practically relevant problem, finds the proposed method empirically effective, and recognizes the paper as providing a structured foundation for sample-specific modality acquisition under acquisition costs. We also appreciate the reviewer's concerns regarding the interpretation of the value functions, the role of the submodularity assumption, the experimental baselines, and the clarity of the formal presentation.
>
> In the revised manuscript, we made two major additions that directly address the reviewer's central concerns: (i) an empirical validation of the submodularity assumption for the learned value models across all experimental settings, and (ii) a new scalable adaptive baseline that separates the benefit of sample-specific acquisition from the benefit of our learned value-based acquisition policy. We also revised the discussion of value-function interpretation, acquisition-cost calibration, training cost, and notation. We address each concern in detail below.
>
> __Weakness 1: Validity and interpretation of the value function.__
>
> We agree with the reviewer that the interpretation of the value functions should be stated more explicitly. In the revised manuscript, we clarified that our value functions are task-defined acquisition rewards, rather than intrinsic or universal measures of modality importance. In our setting, the purpose of the value function is to quantify the expected benefit of acquiring additional modalities for the downstream prediction task, conditioned on the modalities already observed.
>
> Although the realized supervision targets for $Q^A$, $Q^B$, and $Q^C$ are discrete, the learned value model estimates their conditional expectation, which is continuous and directly interpretable. In particular, for our main value function $Q^A$, the Bayes-optimal value model recovers the expected change in prediction accuracy induced by acquiring the candidate modalities. We prove this result in Appendix A, Lemma 3, and now reference this interpretation explicitly in Section 3.4. Thus, a prediction such as $\widehat{Q}=0.07$ should be interpreted as an estimated expected accuracy improvement, not as an abstract latent utility of a modality. This design choice is intentional. We also experimented with richer continuous alternatives, including an information-gain-based value function, the results for which we include in Appendix B. However, we found that these targets were substantially noisier and higher variance in practice, making the corresponding regression problem difficult to learn reliably. We now include these results in the appendix. This motivated our use of transition-based value functions, which provide stable and interpretable supervision while still yielding a continuous expected acquisition value through the learned model. We have revised the manuscript to make this interpretation clearer and to avoid suggesting that the value model estimates a generic modality utility.

---

> > ### Author Response · Authors · 2026-06-17
> > **Part 2**
> >
> > Continuing our response:
> >
> > __Weaknesses 2 and 3: Submodularity of the learned value functions.__
> >
> > We agree with the reviewer that the connection between the submodularity assumption and the learned value models should be made more explicit. This was one of the main additions in the revised manuscript. This was one of the main additions in the revised manuscript. In particular, we added a new empirical validation of the submodularity assumption in Appendix C, and we also updated the discussion in Section 3.2 to summarize these findings. Our optimization procedure in Algorithm 1 relies on the learned value model $\widehat{Q}_f(\cdot;\theta)$ being submodular with respect to the proposed subset of modalities to acquire.  To directly test this property, we evaluate the second-order submodularity condition for the final learned value models across all experimental settings. Specifically, for context modality $\mathbf{c}$, observed modality subset $p$, proposed subset $q$, and modalities $i,j$ not already included, we compute
> >
> > $$\Delta_{ij}(\mathbf{c},\mathbf{X},q,p) = \widehat{Q}_f(\mathbf{c}, \mathbf{X}^{p},q\cup\{i,j\};\theta) - \widehat{Q}_f(\mathbf{c}, \mathbf{X}^{p},q\cup\{i\};\theta) - \widehat{Q}_f(\mathbf{c}, \mathbf{X}^{p},q\cup\{j\};\theta) + \widehat{Q}_f(\mathbf{c}, \mathbf{X}^{p},q;\theta).$$
> >
> > Submodularity requires $\Delta_{ij} \leq 0$, corresponding to diminishing marginal returns. We estimate this quantity empirically over the test set by randomly sampling subsets $p$ and $q$ as in the value-model training procedure, and report the average second-order value $\bar{\Delta}$, its $95\%$ confidence interval, the positive violation rate $PVR_{1\sigma}$, and the statistically significant positive violation rate $SPVR_{0.05}$ after Benjamini--Hochberg correction across modality pairs.
> >
> > The results are summarized below:
> > | Task | $\bar{\Delta}$ | $\bar{\Delta}\,[\ell_{0.95},u_{0.95}]$ | $\mathrm{PVR}_{1\sigma}$ (%) | $\mathrm{SPVR}_{0.05}$ (%) |
> > |---|---:|---:|---:|---:|
> > | PTB-XL EKG | -0.0026 | [-0.0032, -0.0022] | 0.59 | 0.0 |
> > | Patch-MNIST | -0.0037 | [-0.0041, -0.0032] | 0.35 | 0.0 |
> > | MIMIC-III | -0.0313 | [-0.0335, -0.0290] | 4.5 | 0.0 |
> > | CMU-MOSEI | -0.0036 | [-0.0044, -0.0029] | 2.5 | 0.0 |
> >
> > Across all experiments, the average second-order condition is negative, and the upper bound of the $95\%$ confidence interval remains below zero. Thus, in expectation, the learned value models behave as submodular functions across all domains considered. Importantly, we found zero statistically significant positive violations for any modality pair across all experiments. While there are some instance-level positive violations, these do not correspond to a significant pattern of positive violations for any pair of modalities. For example, in MIMIC-III, despite a $PVR_{1\sigma}$ of $4.5$%, the largest pairwise test statistic was still strongly negative, indicating that these occasional positive values do not contradict the overall diminishing-returns structure of the learned value model.
> >
> > We believe this addition directly addresses the reviewer's concern. The original manuscript stated the submodularity assumption and developed the corresponding optimization procedure, but did not empirically verify that the learned value models used in the experiments satisfied this condition. The revised manuscript now provides direct empirical evidence that the relevant learned functions satisfy the required diminishing-returns property in the experimental settings considered. We have also revised the surrounding text to clarify that submodularity is an assumption of the optimization framework, not a property guaranteed for arbitrary multimodal tasks, and that our experiments explicitly validate this assumption for the value models optimized in practice.

---

> > > ### Author Response · Authors · 2026-06-17
> > > **Part 3**
> > >
> > > Continuing our response:
> > >
> > > __Weakness 4: Training cost of the value estimator.__
> > >
> > > We agree with the reviewer that the cost of training the value estimator is important for understanding the practical efficiency of the proposed framework. In the revised manuscript, we added the hardware specifications used for all experiments in Section 4 and now report the wall-clock time required to train each value model, including both seconds per epoch and the total number of epochs for each experiment. We also report value-model training curves, measured by validation AUC, in Appendix B. We clarified that supervision for the value model is generated by sampling modality-subset pairs as described in Algorithm 2. In each epoch, we sample one pair of observed and proposed modality subsets for each sample in the training dataset. Thus, the number of epochs and wall-clock training time reported in Section 4 describe the amount of supervision and training overhead used for the value models in our experiments.
> > >
> > > Across all experiments, the total value-model training time is less than 10 hours, and in some settings is substantially smaller; for example, the MIMIC-III value model trains in approximately 12 minutes. These experiments were run on a single-node setup with 40 Intel Xeon Gold CPU cores and one NVIDIA V100 GPU with 32GB of VRAM. Thus, the reported training times do not rely on large-scale compute infrastructure, and could likely be further reduced with more recent GPUs or multi-GPU parallelism. We also clarified the distinction between the offline value-learning cost and the online acquisition cost. The value model is trained once, offline, using sampled modality subsets and evaluations of the underlying predictor. After this one-time training stage, the learned value model supports sample-specific adaptive acquisition at inference time with millisecond-scale evaluation. Thus, the training cost is amortized over deployment, while the resulting acquisition policy provides high-quality adaptive solutions to the AMA problem without requiring expensive subset search or retraining at inference time.
> > >
> > > __Weakness 5: Treatment of acquisition costs and interpretation of $\lambda$.__
> > >
> > > We agree with the reviewer that the value-cost tradeoff should be clearly interpretable for practitioners. We clarified this point in Section 3.4. Our framework assumes that acquisition costs are represented as scalar quantities, as is standard in cost-sensitive prediction and modality acquisition. These costs may be directly specified in application-relevant units, such as dollars, time, or other practitioner-defined cost scores, or scalarized from multiple considerations according to deployment-specific priorities. Importantly, the interpretation of the value-cost tradeoff is one reason we use the proposed accuracy-change value function. Although the realized target for $Q^A$ is discrete, the learned value model estimates its conditional expectation, which has a direct interpretation as expected change in prediction accuracy. Therefore, once costs are represented in scalar units, the parameter $\lambda$ has a clear operational meaning: it is the exchange rate between expected accuracy gain and acquisition cost. For example, if a modality corresponds to a medical test whose cost is measured in dollars, then $\lambda$ specifies how much additional monetary cost the practitioner is willing to incur for a given expected increase in accuracy, or equivalently how much expected accuracy improvement is required to justify an additional dollar of acquisition cost.
> > >
> > > This interpretation is more transparent than the corresponding tradeoff for a more abstract continuous target, such as information gain, whose units are less directly tied to the final prediction objective. Thus, while our framework still requires practitioners to specify or scalarize deployment costs, the benefit side of the tradeoff is expressed in an interpretable task-level quantity, expected accuracy change.

---

> > > > ### Author Response · Authors · 2026-06-17
> > > > **Part 4**
> > > >
> > > > Continuing our response:
> > > >
> > > > __Weakness 6: Experimental baselines and isolating the contribution of the learned adaptive policy.__
> > > >
> > > > We agree with the reviewer that comparing only against a static modality-selection baseline does not fully isolate the contribution of our learned adaptive acquisition policy. To address this concern, we added a new adaptive baseline in Section 4. Specifically, we extend the Efficient Modality Selection (EMS) baseline to obtain a cost-adaptive EMS policy. The original EMS procedure constructs a greedy sequence of modality subsets $s_1,\ldots,s_M$ using population-level accuracy gains. In the static EMS baseline, for each value of $\lambda$, we select the single subset in this sequence that maximizes the average reward. In the new adaptive EMS baseline, the policy instead adapts to the individual acquisition costs of each sample. Given sample-specific costs $\mathbf{z}_i$, the adaptive EMS policy selects the subset index $k$ maximizing $a(\pi_k) - \lambda \mathbf{z}_i^\top \mathbf{s}_k$ over $k \in [M]$, where $\mathbf{s}_k$ is the binary representation of the $k$-th greedy EMS subset. Thus, unlike static EMS, this baseline can acquire more modalities for samples with unusually low costs and fewer modalities for samples with unusually high costs.
> > > >
> > > > This baseline directly addresses the reviewer's concern because it is sample-level adaptive, scalable, and cost-aware. At the same time, it does not use the currently observed modalities or context to estimate the sample-specific benefit of acquiring additional modalities. Therefore, the comparison separates the benefit of EAMA from adaptivity alone: adaptive EMS tests whether sample-specific cost adaptation is sufficient, while EAMA additionally reasons about the observed information and the expected value of acquiring new modalities. While the reviewer suggested random sample-level baselines with matched acquisition budgets as one possible diagnostic, we instead added a stronger cost-adaptive EMS baseline. This baseline is sample-specific and cost-aware, but does not use the observed modalities or learned value estimates. Thus, it more directly isolates the contribution of our context-aware value model while remaining a principled extension of the strongest scalable baseline.
> > > >
> > > > The results in Section 4 show that adaptive EMS consistently improves over static EMS, as expected, but still underperforms EAMA. In some settings, EAMA achieves up to $6\times$ the average reward of adaptive EMS. These results strengthen the experimental evidence for our method: the gains are not explained solely by moving from static to adaptive acquisition, but by the learned context-aware value model and scalable acquisition procedure used by EAMA.
> > > >
> > > > __Weakness 7: Notation and formal problem setup.__
> > > >
> > > > We thank the reviewer for pointing out these notation issues. We agree that several aspects of the original formulation could be made clearer, even though they do not change the method or experimental setup. In the revised manuscript, we updated Section 3.1 to make the notation more consistent and to clarify the probabilistic interpretation of the value function. First, we now use $[M]$ consistently to denote the set of modality indices, with $M$ reserved for the number of modalities. Thus, observed and candidate modality subsets are written as subsets of $[M]$. We also clarified that, for a fixed instance, $p \subseteq [M]$ denotes the currently observed modalities and $q \subseteq [M]\setminus p$ denotes the candidate modalities being considered for acquisition. Second, we added an explicit note that we suppress the instance index when the context is clear. The problem is still sample-level adaptive modality acquisition: for instance $i$, $\mathbf{X}_i^p$ denotes the subset of modalities observed for that sample. For notational simplicity, the manuscript often writes $\mathbf{X}^p$ when discussing a fixed instance. Third, we clarified the conditional distribution appearing in the value function. The data-generating process is defined over complete multimodal samples $(\mathbf{X},y)\sim\mathcal{D}$, where $\mathbf{X}=(\mathbf{X}^1,\ldots,\mathbf{X}^M)$. This complete-data distribution induces the conditional distribution over the target and unobserved modalities given the observed modalities, e.g., $\mathcal{D}(y,\mathbf{X}^q \mid c,\mathbf{X}^p)$, or $\mathcal{D}(y,\mathbf{X}^q \mid \mathbf{X}^p)$ when no additional context $\mathbf{c}$ is used. The value function is the expected acquisition benefit under this induced conditional distribution. At inference time, $y$ and $\mathbf{X}^q$ are unknown, and the learned value model estimates this conditional expectation from the observed modalities and context. These revisions have helped remove ambiguity in the formal setup.

---

> > > > ### Author Response · Authors · 2026-06-17
> > > > **Part 5**
> > > >
> > > > Continuing our response:
> > > >
> > > > __Requested changes.__
> > > >
> > > > The requested changes largely overlap with the concerns addressed above. In summary, we have: (i) clarified the interpretation of the value functions and the meaning of the learned continuous value estimates; (ii) added an empirical validation of the submodularity assumption for the learned value models; (iii) added a scalable cost-adaptive EMS baseline to better isolate the contribution of EAMA's learned context-aware acquisition policy; (iv) reported value-model training times, hardware specifications, validation AUC curves, and the supervision procedure used to train the value model; (v) clarified the interpretation of acquisition costs and the role of $\lambda$; and (vi) revised the notation and probabilistic formulation in Section 3.1. We believe these revisions directly address the reviewer's requested changes and improve the rigor and clarity of the manuscript. We thank the reviewer again for their comments and feedback.

---

> > ### Comment · Reviewer_C1nG · 2026-06-19
> > **Thank you for the response**
> >
> > I thank the authors for the detailed response and substantial revision. Overall, I appreciate the effort that went into addressing the concerns raised in my review, and I believe the revised manuscript is significantly stronger than the original version.
> > Regarding the interpretation of the value function, I still think there is a subtle distinction between estimating the expected accuracy gain from acquiring a modality and estimating the intrinsic importance of that modality for the task. In practice, the observed accuracy improvement may reflect both the task relevance of a modality and dataset- or environment-specific factors such as data quality. For example, if a sensor performs poorly in one deployment region, the learned acquisition policy may partially reflect local data quality rather than the underlying utility of that modality, which could affect generalization across environments. That said, I think the authors have clarified the intended interpretation sufficiently, and this is no longer a major concern for me.
> > As for the submodularity assumption, I really appreciate that the authors performed a dedicated empirical validation and reported violation rates across datasets. While this is not a theoretical guarantee, it provides meaningful empirical support for the assumption underlying the optimization framework. For my purposes, this addresses the concern reasonably well.
> > For the training cost of the value estimator, my original question about whether learning the value model could potentially cost more than acquiring additional modalities was not fully resolved. However, the newly reported training times are helpful. Seeing that all reported settings require < 10 hours makes the practical overhead appear much more reasonable, and I find this evidence largely convincing.
> > For the treatment of acquisition costs, I still view the reduction of heterogeneous costs to a single deployment specific scalar as more of a convention than a complete solution. Difficult tradeoffs, such as balancing monetary cost against risk or other non-monetary considerations, remain inherently challenging. However, I also recognize that solving this broader problem is beyond the scope of the present paper, and I am satisfied with the authors' clarification of their assumptions.
> > Overall, I appreciate the authors' thoughtful and constructive response. They have addressed the vast majority of my concerns, and I thank them for taking the review seriously and providing detailed revisions and additional analyses.

---

### Review · Reviewer_Spre · 2026-07-14

**Summary Of Contributions:**

This paper proposes Efficient Adaptive Modality Acquisition (EAMA) for Multi-Modal Learning. The main idea is to gradually add necessary modalities in the inference time.

**Audience:**

No

**Audience Explanation:**

With the current form of paper presentation and experimental results, I doubt about the interest of TMLR's audience to the findings of this paper.

**Claims And Evidence:**

No

**Claims Explanation:**

- The writing of this paper is not good with many confusing points, making it hard to understand completely.

- Moreover, it is unclear why not to use all modalities in training and inference. The motivation of gradually selecting more and more modalities in inference is still unclear.

- The idea is heuristic and the connections to RUSM and Greedy in Algorithm 1 are unclear and vague.

- The training pipeline is not clear and coherent to the inference. It is still unclear what models need to be trained: $Q_f$, $\bar{Q}_f$, or $f$. It is also unclear the principle and the objective function of training. It seems to be mentioned in Algorithm 2 (cf. take gradient step on ...), but no further explanations.

- In experiments, there are no report on the metrics such as accuracy.

**Requested Changes:**

- Writing a better story to convince audiences the advantages and benefits of this research.

- Writing paragraphs to summarise the main steps of the proposed algorithm and their motivations and rationales. In the current form, this prevents readers to fully or completely understand the paper.

- Why inference time, we do not use all modalities?

- Report more metrics for experiments.

---

### Author Response · Authors · 2026-06-17

We thank the reviewers for their thoughtful and constructive feedback. We have uploaded a revised manuscript addressing the main points raised across the reviews. The most substantial changes are: (i) an empirical validation of the submodularity assumption for the learned value models across all experiments; (ii) a new cost-adaptive EMS baseline that better isolates the contribution of EAMA's context-aware value model; (iii) additional reporting of value-model training cost, hardware specifications, and validation curves; (iv) clarification of the value-function interpretation, acquisition-cost tradeoff, and MCR-based value-function selection guideline; and (v) improved notation and additional appendix analyses on calibration and learned modality embeddings. We provide detailed responses to each reviewer in the corresponding review threads.